# CaliDist: Calibrating Large Language Models via Behavioral Robustness to Distraction

**Mohammad Anas Jawad** [1]   **Cornelia Caragea** [1]

## Abstract

Existing calibration methods for Large Language Models (LLMs) often overlook a critical dimension of trustworthiness: a model's *behavioral robustness* to irrelevant or misleading information. In this paper, we argue that a model's true confidence should reflect its stability under cognitive pressure. We introduce CALIDIST, a novel post-hoc calibration approach that directly measures and penalizes a model's susceptibility to distraction. CALIDIST quantifies how an LLM's predictions and uncertainty change when its input prompt is perturbed with semantic *distractors*. This stability (or lack thereof) signal is then used to adaptively scale the model's initial confidence score. Our extensive experiments on seven Natural Language Understanding classification benchmarks using six distinct LLMs show that CALIDIST consistently achieves lower Expected Calibration Error (ECE) and Brier Score compared with strong baselines. Remarkably, our method reduces the ECE from 23% to 7% on average—a relative improvement of 70%—demonstrating that behavioral stability is a powerful signal for calibration. We make our code and datasets available at github.com/m-anas-j/CaliDist.

## 1. Introduction

Large Language Models (LLMs) have demonstrated remarkable capabilities across a vast spectrum of complex tasks, leading to their rapid integration into high-stakes applications (Thirunavukarasu et al., 2023). In these applications, the correctness of a model's output is paramount, but equally important is the reliability of its confidence. A well-calibrated model expresses confidence that accurately reflects the true likelihood of its correctness and is essential for building trustworthy systems, enabling their safe adoption (Sun et al., 2024). However, modern LLMs, particularly those fine-tuned with Reinforcement Learning from Human Feedback (RLHF), are often severely miscalibrated, typically exhibiting a strong tendency towards overconfidence (Achiam et al., 2023; Tian et al., 2023).

Existing research on model calibration has largely followed two main paradigms. The first, inherited from traditional deep learning, involves post-hoc statistical adjustments, such as temperature scaling (Guo et al., 2017). While effective, these calibration methods are limited by their requirement for direct access to model logits, rendering them inapplicable to many proprietary LLMs. The second paradigm, developed for the generative nature of LLMs, estimates confidence by measuring response consistency, most notably through self-consistency (Wang et al., 2023). These methods cleverly probe the model's internal uncertainty but often incur significant computational costs. More importantly, they measure consistency against the model's own internal stochasticity, not against external challenges.

This leaves a critical dimension of trustworthiness underexplored: *a model's behavioral robustness*. Recent studies have shown that LLMs are notoriously brittle, often changing their predictions when presented with irrelevant but plausible "distractor" information (Shi et al., 2023). This brittleness mirrors well-documented phenomena in human cognitive psychology, and motivates us to design a principled, behavior-centric view of calibration. Specifically, we can draw two parallels, each corresponding to a distinct, measurable behavior. First, a model's tendency to alter its original prediction after being exposed to misleading information is a computational parallel to the *Misinformation Effect* (Loftus & Palmer, 1974), where human memory of an event can be fundamentally altered by misleading post-event information. Here, the model's initial prediction acts as its "memory" of an answer, and a semantic distractor serves as post-event information that can erroneously corrupt that memory. A model highly susceptible to this effect is inherently less reliable, as it reveals a fundamental flaw in its reasoning process: it indicates that the model's conclusions are not based on a robust, internal understanding of the problem, but are instead heavily influenced by superficial cues in

---

[1]Computer Science, University of Illinois Chicago, USA. Correspondence to: Mohammad Anas Jawad <mjawad4@uic.edu>, Cornelia Caragea <cornelia@uic.edu>.

*Proceedings of the 43rd International Conference on Machine Learning*, Seoul, South Korea. PMLR 306, 2026. Copyright 2026 by the author(s).

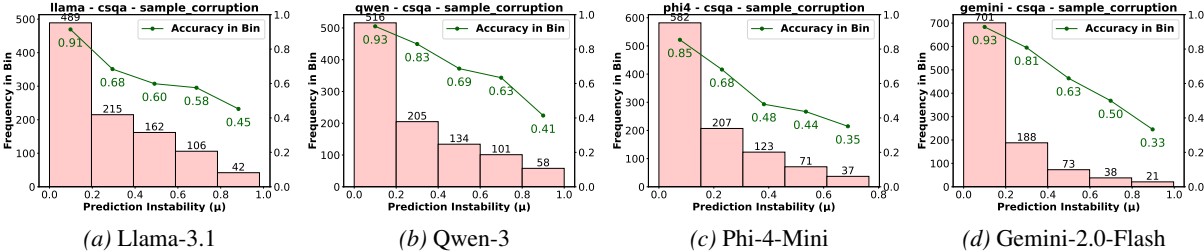

*Figure 1.* Negative correlation between prediction accuracy and prediction instability. Accuracy drops as Prediction Instability $\mu$ increases. Samples on which the models demonstrate higher $\mu$ tend to have lower average accuracy. Distractor used: *Sample-corruption* style.

the prompt. Second, a model that remains highly confident while being easily distracted exhibits a behavior akin to the *Dunning-Kruger Effect*, where low competence on a task is paired with an inflated overestimation of ability (Kruger & Dunning, 1999).[1] In this parallel, the model's "incompetence" is its inability to resist distraction, and its "overestimation" is its failure to reduce its confidence accordingly.

To operationalize these parallels, we introduce CALIDIST, a novel post-hoc approach that quantifies and aggregates these behavioral signals for calibration. CALIDIST systematically perturbs an input prompt with targeted distractors and uses the combined instability signal from both prediction changes and confidence shifts to adaptively scale the model's original confidence score. CALIDIST measures the Misinformation Effect parallel with Prediction Instability ($\mu$), which quantifies how frequently a model's prediction changes when perturbed. CALIDIST also captures the Dunning-Kruger parallel by observing this instability in conjunction with the model's Confidence Stability ($\delta$), or its change in predictive uncertainty.

Crucially, we validate that these behavioral metrics are not just quirks but are directly linked to correctness. As visualized in Figure 1, we observe a strong and consistent negative correlation between Prediction Instability ($\mu$) and the model's accuracy on the original, unmodified prompts. We tested three open-source and one proprietary model and observed that samples exhibiting low instability (e.g., $\mu \leq 0.2$) have a very high average accuracy (often above 80-90%), whereas accuracy drops sharply as instability increases. This finding provides the foundational evidence for our work: susceptibility to distractors is a powerful, measurable proxy for a model's likelihood of error. Because our approach can operate on log-probabilities or verbalized confidences (Tian et al., 2023; Xiong et al., 2024), it is broadly applicable to both white-box and black-box LLMs.

Our main contributions are as follows: (1) We introduce behavioral robustness to distraction as a new and critical dimension for LLM calibration, grounding it in established principles from cognitive psychology, such as the Misinformation Effect and the Dunning-Kruger Effect. We empirically demonstrate its strong correlation with prediction accuracy; (2) We propose CALIDIST, a novel, accuracy-preserving post-hoc calibration approach that quantifies a model's stability (or lack thereof) against semantic distractors to adaptively adjust per-sample confidence scores; (3) We demonstrate the versatility of CALIDIST, showing its effectiveness for both white-box and black-box LLMs; (4) We provide extensive empirical evidence showing that our method significantly reduces Expected Calibration Error (ECE) and Brier Score (BS) across multiple datasets and LLMs and outperforms strong baselines.

## 2. Related Work

**Post-Hoc and Black-Box Model Calibration.** Post-hoc calibration remaps a model's output probabilities without altering its weights. The most common method, Temperature Scaling (TS) (Guo et al., 2017), divides the logits by a learnable scalar $T$, but like other foundational methods such as Platt Scaling (Platt et al., 1999) and Isotonic Regression (Zadrozny & Elkan, 2002), it requires access to model logits. Recent works have extended these ideas to LLMs, for instance, by learning task-specific temperatures (Shen et al., 2024) or adapting TS for semantic-level confidence (Lamb et al., 2025). The primary limitation of these approaches is their inapplicability to black-box APIs. The challenge of calibrating black-box models has spurred research into methods that do not require logit access. One prominent direction is verbalized confidence, where the model is prompted to state its certainty directly (Xiong et al., 2024). These scores can provide a stronger signal than the conditional probabilities of RLHF-tuned models (Tian et al., 2023; Xiong et al., 2024). Other approaches involve training auxiliary models to predict correctness (Ulmer et al., 2024; Pedapati et al., 2024), or using conformal prediction (Azaria & Mitchell, 2023). Contemporary work has also explored probing internal representations. For instance, CCPS (Khanmohammadi

---

[1]From a computational perspective, this behavior corresponds to an inaccurate estimate of *epistemic uncertainty*. We propose that susceptibility to distraction is a measurable proxy for this uncertainty: a model with robust knowledge (low epistemic uncertainty) should not be swayed by irrelevant distractors. Thus, high instability coupled with high confidence signals uncaptured epistemic uncertainty.

et al., 2025) calibrates confidence by analyzing the stability of internal representations against Jacobian-based adversarial perturbations. However, CCPS relies on access to hidden states and requires training extensive auxiliary encoders. In contrast, CALIDIST is a purely behavioral robustness based approach. It requires no access to model internals—neither logits nor hidden states, making it deployable not only on open-source but also on proprietary models (e.g., GPT-4o, Gemini) where methods like CCPS or TS are inapplicable. Thus, our method is distinct as it requires no external model training, deriving its signal directly from the target model's behavior. CALIDIST is fully compatible with log-probabilities or verbalized confidence, effectively acting as a behavioral proxy for TS and achieves a similar confidence-scaling effect without needing logit access.

**Consistency and Perturbation-Based Confidence Estimation.** A separate family of methods estimates confidence by measuring the consistency of a model's outputs across multiple forward passes (Wang et al., 2023; Manakul et al., 2023; Xiong et al., 2024; Wightman et al., 2023). These techniques, which can measure semantic similarity (Lamb et al., 2025) or aggregate votes from paraphrased prompts (Kadavath et al., 2022), often incur substantial computational overhead, typically requiring 10-40 passes for a stable signal (Manakul et al., 2023). SPUQ (Gao et al., 2024) extends self-consistency by introducing non-adversarial perturbations (e.g., paraphrasing, temperature changes) to probe epistemic and aleatoric uncertainty. Similarly, SteerConf (Zhou et al., 2025) manipulates the metacognitive instructions in the prompt (e.g., "be cautious" vs. "be confident") to measure the coherence of the model's self-assessment. CALIDIST differs fundamentally from these approaches in both *objective* and *mechanism*. While SPUQ measures stability against semantic equivalents and SteerConf measures stability against instructional steering, CALIDIST measures behavioral robustness against adversarial misinformation. We do not ask the model to be cautious or rephrase the input; we actively inject misleading "distractors" to test if the model's reasoning path is robust to the Misinformation Effect. Additionally, unlike SPUQ or SteerConf, which aggregate multiple outputs into a new confidence score, CALIDIST explicitly rescales the initial confidence score using a learned parametric scaling factor. This allows CALIDIST to function as a sample-specific behavioral proxy for Temperature Scaling, a property we empirically validate in §5.

**Bayesian and Parameter-Space Approaches.** A distinct line of research models uncertainty through the lens of Bayesian inference. Methods such as Bayesian Low-Rank Adaptation (Yang et al., 2024), Blob (Wang et al., 2024), and Information-Theoretic Evidential Deep Learning (Li et al., 2025) estimate uncertainty by modeling the posterior distribution of model parameters (e.g., LoRA weights). While theoretically grounded, these approaches operate in the parameter space, requiring access to weights and are often computationally expensive. CALIDIST, conversely, operates in the behavioral space at inference time. It is a lightweight, post-hoc method that requires no training, making it compatible with frozen, API-based models.

**LLM Robustness to Adversarial Context.** A parallel stream of research has established that LLMs are brittle and easily swayed by irrelevant context or distractors (Shi et al., 2023; Chen et al., 2024; Mozes et al., 2023; Wang et al., 2025). Xiong et al. (2024) employ induced consistency to sample multiple responses. While existing work uses these failures to demonstrate LLM limitations, to our knowledge, our work is the first to formally bridge the concept of distraction robustness with confidence calibration, leveraging this behavioral signal as a core component of our framework.

## 3. Methodology

In this section, we elaborate on the working principle of the CALIDIST framework. Our proposed framework calibrates the confidence of an LLM by evaluating its behavioral robustness. Instead of relying solely on a model's initial, static confidence score, our method leverages the stability (or lack thereof) of its predictions and certainty when presented with "distractor" information. The core intuition is that a reliable model should remain stable in both its conclusion and its associated confidence, even under cognitive load. We show the operational flow of CALIDIST in Figure 2. The algorithm is presented in Appendix A.

### 3.1. Formalism and Notation

Let $M$ be the model, $\pi_o$ be the original input prompt, $\mathbf{x}$ be the input, $y$ be the ground truth, and $\hat{y}$ be the model's initial prediction, for which it produces an initial confidence score $p = P(\hat{y}|\pi_o(\mathbf{x}))$. This confidence can be derived from either the logit-based probability of the output token (for open-source models) or from the log-probability and verbalized confidence (for both open-source and black-box models). We define a set of $k$ distractors $D = \{d_1, d_2, \ldots, d_k\}$. Each distractor $d_j$ is a piece of information designed to be semantically related to the task but logically irrelevant or contradictory to the initially predicted reasoning path for the input $\mathbf{x}$. For each distractor $d_j$, we construct a new, distracted prompt $\pi_{d_j} \leftarrow \pi_o \oplus d_j$, where $\oplus$ denotes the concatenation or integration of the distractor into the original prompt. The model's forward pass on this new prompt yields a new prediction $\tilde{y}_{d_j}$ with a corresponding confidence score $p'_j = P(\tilde{y}_{d_j}|\pi_{d_j}(\mathbf{x}))$.

### 3.2. Distractor Styles

We design three distinct styles of distractors to induce different types of perturbations in the responses of LLMs: *Assertion-style*, *Probe-style*, and *Sample-Corruption-style*.

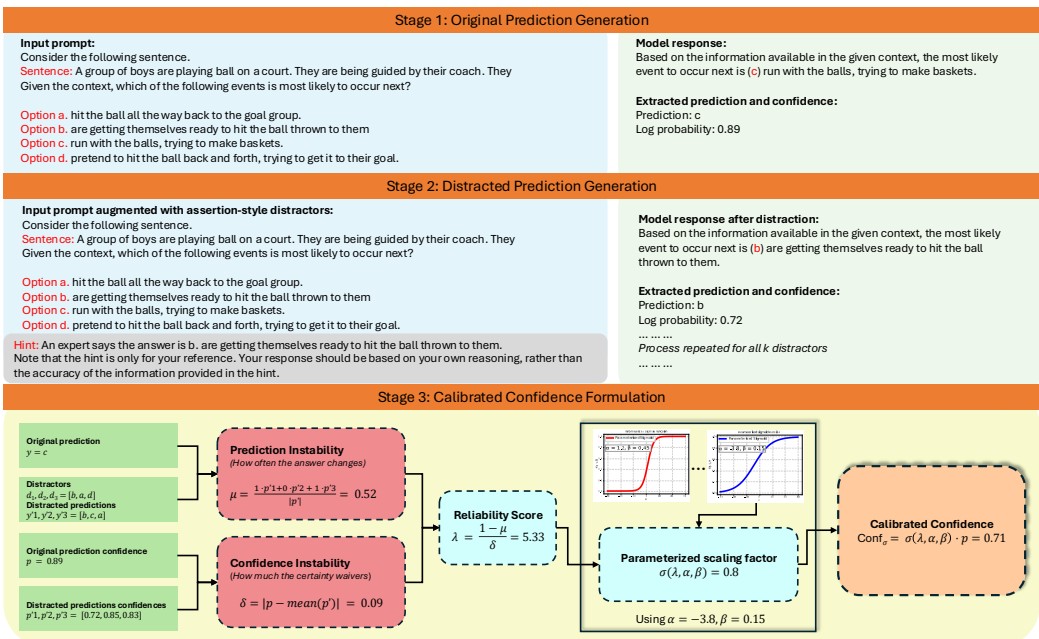

*Figure 2.* Illustration of the CALIDIST framework. Note that in experiments we use multiple distractor prompts per example; here, for simplicity, we show only one distractor prompt.

**Assertion-style Distractors.** Assertion-style distractors are authoritative assertions appended to the original prompt to try to deviate a model's initial response. For example, an assertion-style distractor may state, "*Wikipedia claims the answer is . . .*", followed by a misdirecting label (i.e., a label that is different from the original prediction $\hat{y}$).

**Probe-style Distractors.** These distractors are designed to encourage the model to consider alternative answers, thereby inducing uncertainty and checking for deviation from its original response. An example of a probe-style distractor is, "*Do you think the answer could be . . .?*"

**Sample-Corruption-style Distractors.** This style introduces a disruptive paradigm by corrupting the samples themselves with misleading cues. For instance, in the MNLI benchmark (Williams et al., 2018), if the premise is "*She is known to many as the Queen of Rockabilly*" and the hypothesis is "*She is known to many as the Queen of Rockabilly or the First Lady of Rockabilly*", the hypothesis can be corrupted by adding "*This sentence contradicts the premise.*" as a distractor. The primary goal of this distractor style is to observe any deviations in the model's responses caused by distractors embedded in the sample and to use these deviations as calibration signals.

The formulation of these distractor-styles is motivated by well-documented psychological phenomena discussed in Section 1—specifically, the Misinformation Effect and the Dunning-Kruger Effect—allowing us to systematically operationalize distinct human cognitive failure modes into

concrete engineering metrics. We emphasize that CALIDIST is entirely distractor-agnostic by design; any structured perturbation capable of eliciting a measurable behavioral change in the model serves as a valid calibration signal. Furthermore, these distractors are generated automatically at runtime via a handful of predefined semantic templates used in conjunction with the task's label space (detailed in Algorithm 1 and Appendix E and F). This design choice establishes a crucial dichotomy between system setup and end-user experience: the system designer defines the overarching task parameters and the distractor templates once at the initialization phase, leaving the end user completely unburdened. The end user can interact with the language model as per usual without needing to construct, optimize, or even think about the underlying distractors, all while receiving fundamentally better-calibrated responses.

### 3.3. CALIDIST: Distraction-Based Calibration

CALIDIST follows a systematic, multi-step process for each input sample to derive a final, calibrated confidence score.

**Elicit Initial Prediction.** First, we use the original prompt $\pi_o$ on sample $\mathbf{x}$ to obtain the baseline (original) prediction $\hat{y}$ and its associated confidence $P(\hat{y}|\mathbf{x})$.

**Generate Distractors and Elicit Distracted Predictions.** For a given classification task with $c$ possible labels, we generate $k = (c - 1) \times m$ distractors, where $m$ is a hyperparameter that denotes the number of distractors to generate per class. Each distractor is constructed based on one of the originally unselected labels, creating a set of plausible alter-

native contexts. For instance, in the MNLI task with labels {Entailment, Contradiction, Neutral}, if the original prediction $\hat{y}$ is "Contradiction", we generate distractors related to "Entailment" and "Neutral". This creates a structured and principled set of challenges for the model. We use the $k$ distracted prompts $\pi_{d_1}, \cdots, \pi_{d_k}$ to obtain the prediction $\tilde{y}_{d_j}$ and its confidence $P(\tilde{y}_{d_j}|\pi_{d_j}(\mathbf{x}))$ for all $j = 1, \cdots, k$.

**Quantify Prediction and Confidence Instability.** Using the set of responses from the distracted prompts, we calculate two core metrics of instability:

- **Prediction Instability ($\mu$):** Prediction Instability measures the frequency with which the model changed its prediction from the original answer $\hat{y}$. It is calculated as the weighted confidence scores of the fraction of distracted predictions $\tilde{y}_{d_j}$ that do not match $\hat{y}$:

$$\mu \leftarrow \frac{1}{k} \sum_{j=1}^{k} \mathbb{I}(\tilde{y}_{d_j} \neq \hat{y}) \cdot P(\tilde{y}_{d_j}|\pi_{d_j}(\mathbf{x})) \quad (1)$$

  A high $\mu$ indicates the model is easily swayed by irrelevant information, whereas low $\mu$ indicates stability.

- **Confidence Instability ($\delta$):** This metric measures the magnitude of the shift in the model's average confidence level between its original prediction and its predictions under distraction. It is defined as:

$$\delta \leftarrow \left| P(\hat{y}|\pi_o(\mathbf{x})) - \frac{1}{k} \sum_{j=1}^{k} P(\tilde{y}_{d_j}|\pi_{d_j}(\mathbf{x})) \right| \quad (2)$$

  A large $\delta$ signifies that the model's certainty is volatile and unreliable.

**Calculate Reliability Score ($\lambda$).** We synthesize the two instability metrics into a single scalar reliability score, $\lambda$. This formulation is designed to capture distinct behavioral failure modes:

$$\lambda_{\text{raw}} = \frac{1 - \mu}{\delta + \epsilon} \quad (3)$$

where $\epsilon$ is a small constant (e.g., $1e - 10$) for numerical stability. The interaction between the numerator and denominator is structural, not a simple heuristic choice:

- **$\mu$ as a Gating Mechanism:** Prediction flips represent catastrophic semantic failures. The term $(1 - \mu)$ ensures that as Prediction Instability approaches 1 (total failure), the reliability score $\lambda$ linearly collapses to 0, regardless of the confidence stability.
- **$\delta$ as a Sensitivity Gain:** Confidence fluctuations are subtle, continuous signals. Placing $\delta$ in the denominator acts as a gain amplifier, ensuring the score remains highly sensitive to minor tremors in certainty even when predictions remain stable.

To standardize the dynamic range of the scores and prevent saturation in the subsequent calibration step, we use Min-Max normalization to map $\lambda_{\text{raw}}$ to a fixed interval $[0, 10]$ using a small held-out validation set:

$$\lambda = \text{MinMax}(\lambda_{\text{raw}}) \times 10 \quad (4)$$

**Final Confidence Calibration.** Finally, the normalized reliability score $\lambda$ is mapped to a calibration scalar via a parameterized sigmoid function $\sigma$. This factor scales the original confidence to yield the final calibrated probability score, $\text{Conf}_\sigma$:

$$\text{Conf}_\sigma = \sigma(\lambda, \alpha, \beta) \cdot P(y|x) \quad (5)$$

where

$$\sigma(\lambda, \alpha, \beta) = \frac{1}{1 + e^{-\beta(\lambda - \alpha)}} \quad (6)$$

Crucially, this formulation is not arbitrary; it is grounded in Platt Scaling (Platt et al., 1999). From the perspective of Empirical Risk Minimization (ERM) (McCullagh, 2019), we treat the calibration task as a binary classification problem (predicting correctness) using $\lambda$ as the input feature. The sigmoid function serves as the canonical link function for the Bernoulli distribution, and the sigmoid parameters $\alpha$ (centering) and $\beta$ (scaling) are learned by minimizing the Brier Score, a strictly proper scoring rule, over the same held-out validation set used in $\lambda$ normalization in Equation 4. We provide the full theoretical derivation and proof of equivalence to Platt Scaling in Appendix B.

This formulation balances sample-level granularity with task-level optimization. While $\mu$ and $\delta$ are estimated per sample to capture instance-specific diagnostic signals, $\alpha$ and $\beta$ are hyperparameters that define the shape of the parametric sigmoid function (Generalized Logistic Function) and are estimated per task (or dataset). This design allows CALIDIST to dynamically adapt to different root causes of overconfidence. For instance, sample-level knowledge errors typically manifest as high prediction instability ($\mu$), where the model flips its answer under distraction. Conversely, sample-level reasoning defects manifest as high confidence instability ($\delta$), capturing the 'shaken but stubborn' behavior where the model adheres to its prediction but betrays volatile underlying logic (in Appendix C we provide the intuitive logic of our framework's behavior using three key scenarios). Unifying these signals into $\lambda$ ensures that CALIDIST's corrective penalty scales precisely with individual sample vulnerabilities.

## 4. Experiment Setup

**Datasets.** We evaluate the effectiveness of our proposed CALIDIST approach on seven diverse and challenging benchmarks spanning multiple Natural Language Understanding (NLU) tasks. For Natural Language Inference

(NLI), we use MNLI (Williams et al., 2018), a large-scale, multi-genre corpus, and MSciNLI (Sadat & Caragea, 2024), which specifically tests inference over scientific texts. For paraphrase identification, we include the Twitter PPDB (Lan et al., 2017), a dataset of paraphrase pairs drawn from social media. For Commonsense Reasoning, we employ HellaSwag (Zellers et al., 2019), a sentence completion task that requires predictive reasoning, and Commonsense QA (Talmor et al., 2019) a multiple-choice question-answering benchmark. Finally, to test performance on more complex reasoning and knowledge-intensive tasks, we use Yahoo Answers (Zhang et al., 2015), a large-scale topic classification dataset, and AQuA-RAT ((Ling et al., 2017), a collection of algebraic word problems that require step-by-step rationales. For each dataset, we procure 1000 samples from their respective test sets and a separate set of 200 samples as the held-out validation set (when explicit validation sets are unavailable). This selection of datasets enables us to evaluate our method's performance across a diverse range of domains and reasoning types.

**Models.** Our experiments are conducted on a diverse suite of six state-of-the-art language models to ensure that our findings are broadly applicable. For open-source models, we use four prominent LLMs: Llama-3.1 8B Instruct (Grattafiori et al., 2024), Qwen3 8B (Yang et al., 2025), Phi-4-mini Instruct (Abouelenin et al., 2025), and Gemma-3 4B Instruct (Kamath et al., 2025). These models were selected for their strong performance and varying architectural designs. To validate the effectiveness of our framework in black-box scenarios where logit access is unavailable, we also evaluate two leading proprietary models accessed via their APIs: GPT-4o-Mini (OpenAI, 2024) and Gemini 2.0 Flash (Google, 2024). These models expose token-wise log-probabilities, enabling our evaluation protocol to be applied to a set of different types of confidence values.

**Baselines.** We evaluate the performance of CALIDIST against several well-established baselines to contextualize its effectiveness. Given that our method uses multiple forward passes to assess reliability, our primary comparison is with state-of-the-art consistency-based approaches. We implement three variants: **Self-Consistency** (Wang et al., 2023), which relies on a simple majority vote over stochastic samples; **Entropy-based Consistency**, which uses the entropy of the output distribution as a confidence measure; and **First-Second-Distance Consistency** (FSD), which measures the probability gap between the top two most frequent predictions (Lyu et al., 2025). For all consistency methods, we use a standard of 15 forward passes per sample. Additionally, for open-source models where logits are accessible, we compare our approach with **Temperature Scaling** (TS) (Guo et al., 2017). Since our method can be viewed as a behavioral proxy for TS in black-box settings, this comparison serves as an important point of reference. Finally,

we include the **uncalibrated Vanilla Confidence** (i.e., the model's raw output probability) and **verbalized confidence** (i.e., the verbalized confidence generated as a response) as baselines to quantify the absolute improvement gained by each calibration method. Appendix C provides a detailed description of each baseline.

**Our Method: CALIDIST Variants.** We experiment with two different applications of CALIDIST, based on the distractor style and the confidence value used. For example, CALIDIST (As.) uses the default probability-based confidence with assertion-style distractor, while the verbalized variant is denoted as CALIDIST$_{\text{verbalized}}$ (As.) We provide additional details about our approach in Appendix G.

**Evaluation Metrics.** To evaluate the calibration of our models, we use two standard metrics – Expected Calibration Error (ECE) and Brier Score (BS). Details about these metrics can be found in Appendix H. Note that we do not report accuracy since CALIDIST is an accuracy-preserving method that does not alter the model's original prediction.

**Hyperparameter Tuning.** The parameters $\alpha$ and $\beta$ in the final sigmoid function are critical for tuning the calibration behavior. These are not fixed values but are determined empirically for each task. We use the same held-out validation set (as for Eq. 4) to perform a grid search over a predefined range of values for $\alpha$ and $\beta$. The optimal combination is selected as the one that minimizes Brier Score on this validation set. This ensures that the penalty function is well-suited to the specific task and model being evaluated.

Additional implementation details are in Appendix I.

## 5. Results and Observations

Our experiments, detailed in Tables 1, 2, and 3, confirm the effectiveness and versatility of CALIDIST. We analyze its performance on: 1) open-source models with full logit access; 2) in simulated black-box settings using verbalized confidence; and 3) finally on proprietary, API-based models. We show the results of LLAMA-3.1 and QWEN-3 in Tables 1 and 2. The results for PHI-4-MINI and GEMMA-3 are shown in Appendix J.

**CALIDIST as a Superior Behavioral Proxy for Temperature Scaling.** Table 1 shows the comparison between CaliDist (with all distractor styles) and TS. Our results validate that CALIDIST effectively functions as a behavioral proxy for TS in black-box settings. Crucially, Table 1 demonstrates that CALIDIST frequently outperforms white-box TS. We attribute this empirical advantage to the granularity of the scaling mechanism that CALIDIST provides. This allows our method to selectively penalize specific brittle predictions without uniformly dampening the confidence

*Table 1.* Comparison of CALIDIST with four baselines using two open-source LLMs. Confidence used for all baselines except for Consistency, Entropy, and FSD are logit-based confidence scores. Metrics are given by $\times 10^2$. The best-performing values are in **bold**.

| LLM | Metric | MSciNLI | | MNLI | | PPDB | | Yahoo! Answers | | HellaSwag | | CSQA | | AQuA | | Average | |
|---|---|---|---|---|---|---|---|---|---|---|---|---|---|---|---|---|---|
| | | ECE↓ | BS↓ | ECE↓ | BS↓ | ECE↓ | BS↓ | ECE↓ | BS↓ | ECE↓ | BS↓ | ECE↓ | BS↓ | ECE↓ | BS↓ | ECE↓ | BS↓ |
| **LLAMA-3.1 8B** | Temperature Scaling | 12.10 | 22.38 | 5.83 | 19.27 | 5.12 | 18.98 | 12.64 | 17.72 | **1.98** | **16.87** | 5.57 | 14.73 | 8.71 | 17.11 | 7.42 | 18.15 |
| | Vanilla | 25.76 | 28.30 | 2.81 | 19.01 | 13.50 | 20.87 | 19.74 | 21.18 | 8.66 | 17.65 | 12.59 | 16.57 | 23.18 | 22.82 | 15.18 | 20.91 |
| | Consistency | 34.30 | 34.84 | 11.29 | 21.03 | 19.35 | 23.34 | 25.01 | 24.82 | 17.82 | 20.92 | 17.87 | 19.41 | **4.88** | **14.51** | 18.65 | 22.70 |
| | Entropy | 28.05 | 30.75 | 32.14 | 32.44 | 26.68 | 27.88 | 19.89 | 21.25 | 18.83 | 21.59 | 18.37 | 19.71 | 33.22 | 28.20 | 25.31 | 25.97 |
| | FSD | 29.09 | 29.09 | 17.99 | 24.51 | 23.26 | 24.78 | 21.04 | 22.65 | 16.03 | 20.41 | 15.95 | 18.89 | 20.07 | 19.63 | 20.49 | 22.85 |
| | CaliDist (As.) | 5.75 | 21.94 | **2.74** | 19.01 | 7.02 | 19.16 | **4.11** | **16.88** | 4.24 | 16.92 | 4.73 | **14.70** | **2.02** | 17.65 | 4.37 | 18.04 |
| | CaliDist (Pr.) | **4.98** | 21.82 | 2.83 | 19.02 | **3.75** | 18.91 | 4.56 | 16.89 | 4.17 | 16.99 | 5.32 | 14.77 | 2.98 | 17.45 | **4.08** | **17.98** |
| | CaliDist (Sa.) | 6.23 | **21.58** | 2.77 | 19.01 | 5.42 | **18.72** | 5.85 | 17.20 | **4.11** | 17.02 | **4.32** | 15.01 | 2.73 | **17.25** | 4.49 | 18.11 |
| **QWEN-3 8B** | Temperature Scaling | 35.46 | 35.72 | 27.16 | 28.35 | 38.80 | 39.05 | 28.45 | 28.36 | 15.95 | 16.92 | 14.83 | 15.86 | 8.28 | 10.63 | 24.13 | 24.98 |
| | Vanilla | 38.60 | 38.60 | 29.37 | 29.87 | 39.46 | 39.66 | 29.42 | 29.38 | 18.05 | 18.42 | 16.39 | 16.91 | 12.54 | 12.70 | 26.26 | 26.51 |
| | Consistency | 40.14 | 40.14 | 30.25 | 30.54 | 39.47 | 39.46 | 29.86 | 29.89 | 18.82 | 18.76 | 14.87 | 30.37 | 9.15 | 9.63 | 25.40 | 28.40 |
| | Entropy | 38.23 | 39.07 | 30.65 | 30.89 | 38.96 | 38.87 | 29.05 | 29.21 | 17.12 | 17.83 | 14.86 | 30.40 | 8.90 | 10.86 | 25.40 | 28.16 |
| | FSD | 38.87 | 39.38 | 30.58 | 30.74 | 39.10 | 39.14 | 29.52 | 29.57 | 17.86 | 18.15 | 17.17 | 17.36 | 7.37 | **9.48** | 25.78 | 26.26 |
| | CaliDist (As.) | **2.08** | 22.76 | 6.49 | **21.36** | 5.87 | 21.93 | **3.73** | **18.47** | **3.41** | 14.99 | **1.57** | 12.45 | 3.27 | 11.17 | **3.77** | **17.59** |
| | CaliDist (Pr.) | 6.51 | 23.44 | 11.04 | 22.00 | 4.57 | 24.72 | 5.11 | 18.49 | 3.71 | 14.90 | 2.56 | **12.18** | **2.82** | 11.21 | 5.19 | 18.13 |
| | CaliDist (Sa.) | 2.88 | **22.55** | **6.38** | 21.43 | **2.39** | **21.38** | 6.53 | 19.62 | 3.70 | **14.62** | 4.80 | 13.62 | 3.90 | **11.16** | 4.37 | 17.77 |

*Table 2.* Comparison of CALIDIST with the verbalized confidence method using two open-source LLMs. Metrics are given by $\times 10^2$. The best-performing values are in **bold**.

| LLM | Metric | MNLI | | PPDB | | Yahoo! Answers | | HellaSwag | | AQuA | | Average | |
|---|---|---|---|---|---|---|---|---|---|---|---|---|---|
| | | ECE↓ | BS↓ | ECE↓ | BS↓ | ECE↓ | BS↓ | ECE↓ | BS↓ | ECE↓ | BS↓ | ECE↓ | BS↓ |
| **LLAMA-3.1 8B** | Verbalized | 26.40 | 28.49 | 14.63 | 19.83 | 16.82 | 21.10 | 11.56 | **21.06** | 20.57 | 24.98 | 18.00 | 23.09 |
| | CaliDist$_{\text{verbalized}}$ (As.) | 10.03 | 23.22 | 5.07 | 19.55 | 4.96 | 18.65 | 9.15 | 23.55 | 20.59 | 26.46 | 9.96 | 22.29 |
| | CaliDist$_{\text{verbalized}}$ (Pr.) | **7.00** | **22.37** | 7.70 | 19.37 | **3.51** | **18.47** | **9.11** | 23.54 | **20.16** | 25.04 | **9.50** | 21.76 |
| | CaliDist$_{\text{verbalized}}$ (Sa.) | 10.20 | 23.26 | **2.28** | **17.52** | 5.07 | 18.75 | 9.12 | 23.54 | 20.84 | **24.16** | **9.50** | **21.45** |
| **QWEN-3 8B** | Verbalized | 9.28 | 19.21 | 31.64 | 32.32 | 13.10 | 21.33 | **3.23** | **15.73** | 16.74 | 17.63 | 14.80 | 21.24 |
| | CaliDist$_{\text{verbalized}}$ (As.) | 7.79 | 19.05 | 6.86 | 21.51 | **3.04** | 19.82 | 3.52 | 15.98 | **13.71** | 17.68 | 6.98 | 18.81 |
| | CaliDist$_{\text{verbalized}}$ (Pr.) | 9.18 | 19.10 | 9.34 | 23.86 | 3.29 | **19.71** | 3.52 | 15.98 | 14.44 | 17.83 | 7.95 | 19.30 |
| | CaliDist$_{\text{verbalized}}$ (Sa.) | **7.27** | **18.98** | **2.61** | **20.84** | 3.85 | 19.97 | 3.52 | 15.98 | 14.06 | **17.41** | **6.26** | **18.64** |

of robust samples, effectively providing a higher-resolution signal than global temperature scaling.

**CALIDIST outperforms consistency-based calibration methods.** CALIDIST consistently outperforms all three consistency-based baselines (Consistency, Entropy, FSD). This suggests that measuring a model's stability against external, adversarial distractors is a more effective reliability signal than measuring its internal consistency across stochastic reasoning paths.

**Effectiveness in Verbalized, Black-Box Settings.** To validate CALIDIST's applicability to black-box models, we tested a verbalized confidence variant on the same open-source LLMs. The results in Table 2 are unequivocal: CALIDIST$_{\text{verbalized}}$ achieves a lower ECE than the uncalibrated Verbalized baseline in every single setting. This demonstrates that our approach can significantly enhance reliability using only natural language outputs, without any access to model internal weights.

**Analysis of distractor styles.** While no single variant is universally optimal, we observe strong trends in the performance of our different distractor styles. CALIDIST-Assertion (As.) and CALIDIST-Sample-Corruption (Sa.) emerge as the most consistently effective strategies. These distractor styles tend to highly disrupt the models' initial responses, suggesting that distractor styles with strong disruption tendencies yield better calibration. Nevertheless, the consistent calibration gains achieved across all three distinct

distractor styles in Tables 1 and 2 provide strong empirical evidence of CALIDIST's generalization capabilities across distractor styles. Rather than being overly sensitive to the precise engineering of a single handcrafted prompt, the framework successfully extracts a robust calibration signal from entirely different perturbation types, demonstrating that behavioral instability is a universal proxy for model error regardless of the specific distractor architecture utilized.

**Performance on Proprietary Models.** When leveraging proprietary, API-exposed log-probabilities (Table 3), CALIDIST achieves state-of-the-art calibration on GPT-4o-Mini and Gemini-2.0-Flash, substantially outperforming all baselines. This confirms that our behavioral signals remain highly effective even on the most advanced models. Results are more nuanced when relying purely on verbalized confidence; while CALIDIST still provides notable ECE improvements, the gains are less pronounced compared to the log-probability setting.

## 6. Ablations and Analysis

**Impact of Sigmoid Scaling Parameters ($\alpha, \beta$).** A core component of our method is the parameterized sigmoid function, which translates the reliability score $\lambda$ into the final scaling factor. To justify the necessity of tuning these parameters, we compare the performance of our standard method, which uses optimal $\alpha$ and $\beta$ values found via grid search on the validation set, against a "Default Sigmoid" baseline where $\alpha$ is set to 0 and $\beta$ is set to 1. As shown in Figure 3,

*Table 3.* Performance comparison of CaliDist compared to four baselines using two proprietary LLMs. Metrics are given by $\times 10^2$. The best-performing values for each confidence type except for consistency-based methods are in **bold**.

| Confidence Type | Metric | MSciNLI | | HellaSwag | | CSQA | | Average | |
|---|---|---|---|---|---|---|---|---|---|
| | | ECE↓ | BS↓ | ECE↓ | BS↓ | ECE↓ | BS↓ | ECE↓ | BS↓ |
| **GPT-4O-MINI** | | | | | | | | | |
| *Consistency* | Consistency | 33.20 | 33.28 | 10.90 | 11.99 | 15.52 | 16.07 | 19.87 | 20.45 |
| | Entropy | 25.23 | 29.36 | 11.04 | 12.31 | 15.30 | 16.00 | 17.19 | 19.22 |
| | FSD | 28.76 | 30.77 | 9.37 | 11.51 | 14.95 | 15.82 | 17.69 | 19.37 |
| *Log-Prob.* | Vanilla | 34.97 | 35.14 | 12.96 | 13.14 | 13.87 | 14.21 | 20.60 | 20.83 |
| | CaliDist (As.) | 4.01 | 21.83 | **3.18** | 10.84 | 4.71 | 11.51 | 3.64 | 14.64 |
| | CaliDist (Pr.) | 4.06 | **21.59** | 3.36 | **10.48** | **1.04** | **10.07** | **2.82** | **14.05** |
| | CaliDist (Sa.) | **3.92** | 22.58 | 5.07 | 11.04 | 3.42 | 10.84 | 4.14 | 14.82 |
| *Verbalized* | Verbalized | 13.25 | 24.98 | **13.79** | **14.61** | 8.50 | 13.08 | 11.85 | 17.56 |
| | CaliDist$_{verb.}$ (As.) | 13.59 | **24.53** | 14.38 | 14.61 | 8.59 | **13.03** | 12.19 | **17.39** |
| | CaliDist$_{verb.}$ (Pr.) | 13.96 | 25.73 | 14.83 | 14.26 | 7.62 | 13.38 | 12.14 | 17.79 |
| | CaliDist$_{verb.}$ (Sa.) | **11.47** | 24.89 | **13.79** | 14.59 | **7.55** | 13.27 | **10.94** | 17.58 |
| **GEMINI-2.0-FLASH** | | | | | | | | | |
| *Consistency* | Consistency | 30.28 | 30.75 | 9.23 | 9.39 | 12.97 | 13.21 | 17.49 | 17.78 |
| | Entropy | 28.14 | 29.69 | 8.76 | 9.16 | 12.49 | 13.03 | 16.46 | 17.29 |
| | FSD | 28.09 | 29.59 | 8.79 | 9.11 | 12.18 | 12.80 | 16.35 | 17.17 |
| *Log-Prob.* | Vanilla | 29.81 | 30.38 | 9.88 | 9.61 | 12.82 | 13.08 | 17.50 | 17.69 |
| | CaliDist (As.) | 5.46 | 20.44 | 5.16 | 8.55 | **2.39** | 10.89 | **4.34** | 13.29 |
| | CaliDist (Pr.) | **4.82** | **18.88** | **4.90** | **8.27** | 5.19 | **10.71** | 4.97 | **12.62** |
| | CaliDist (Sa.) | 5.87 | 21.03 | 5.89 | 8.69 | 3.44 | 11.36 | 5.07 | 13.69 |
| *Verbalized* | Verbalized | 4.91 | 21.66 | **22.60** | **13.80** | **10.56** | **13.01** | 12.69 | 16.16 |
| | CaliDist$_{verb.}$ (As.) | 4.89 | 21.09 | **22.60** | **13.80** | **10.56** | **13.01** | 13.00 | 15.92 |
| | CaliDist$_{verb.}$ (Pr.) | 5.04 | 21.27 | **22.60** | **13.80** | **10.56** | **13.01** | 12.73 | 16.03 |
| | CaliDist$_{verb.}$ (Sa.) | **3.75** | **20.79** | **22.60** | **13.80** | **10.56** | **13.01** | **12.30** | **15.87** |

*Table 4.* Computational overhead comparison across calibration methods. Input and output token counts are reported as relative multiples of the baseline.

| Method | Total Forward Passes | Output Tokens (Relative) | Input Tokens (Relative) | ECE↓ |
|---|---|---|---|---|
| Vanilla/Verbalized | 1 | $1 \times 74$ | $1 \times 1064$ | 25.76 |
| Self-Consistency (Entropy) | 15 | $15 \times 74$ | $15 \times 1064$ | 28.05 |
| CaliDist (As.) ($m = 1$) | 3 | $3 \times 98$ | $3 \times 1402$ | 8.02 |
| CaliDist (Sa.) | 3 | $3 \times 65$ | $3 \times 1184$ | 6.14 |

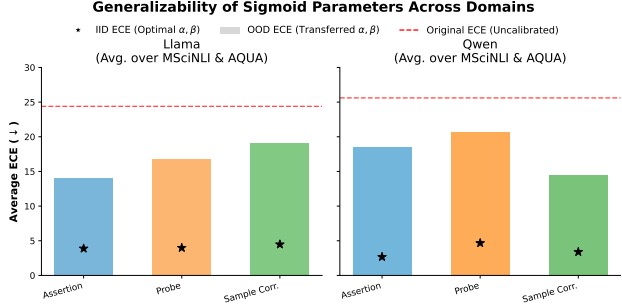

*Figure 4.* Generalizability of $\alpha, \beta$ across domains.

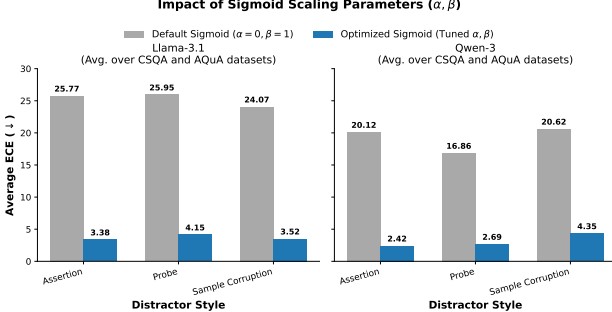

*Figure 3.* Impact of different formulations of Scaling Factor $\sigma$.

for both Llama-3.1 and Qwen-3, the optimized sigmoid consistently and dramatically reduces the ECE compared to the default, untuned version. This demonstrates that learning a task- and model-specific mapping from the reliability score to the final scaling factor is not merely a minor optimization but a critical step for achieving the best possible calibration.

**Computational Efficiency.** A key practical advantage of CaliDist is its computational efficiency compared to consistency-based methods. While techniques like self-consistency often require a large number of forward passes, ranging from 15 to 40 passes to obtain a stable signal (Wang et al., 2023), CaliDist uses a principled and substantially smaller number. The required passes are determined by the task's label space, $k = (c - 1) \cdot m$, where $c$ is the number of classes. Empirically, we find that setting $m = 1$ is sufficient for CaliDist to vastly outperform the consistency baselines. For instance, on our most label-intensive benchmark, Yahoo Answers ($c = 10$), CaliDist with probe-style distractor achieves superior calibration with only 9 distractors (forward passes). In contrast, the baselines remain less

effective even with a higher budget of 15 passes. We perform a separate analysis on the increasing value of $m$ in Appendix J.3 and show in Figure 6 that calibration performance plateaus after a certain threshold. Additionally, to provide a more granular view of computational efficiency, we show the Average Input-Output Tokens per sample for Llama-3.1 on MSciNLI in Table 4. We observe that CaliDist methods require $\approx 5$ times fewer tokens on average, while achieving a calibration improvement of $\approx 75\%$ over the more expensive self-consistency methods, highlighting the efficiency and effectiveness of CaliDist.

**Generalizability of Sigmoid Parameters.** To assess the practical utility of our framework, we tested the generalizability of the learned sigmoid parameters ($\alpha, \beta$) in an out-of-domain (OOD) setting. As shown in Figure 4, parameters tuned on easier source tasks, such as MNLI and CSQA, transfer effectively to more challenging target tasks, such as MSciNLI and AQuA, consistently outperforming the uncalibrated baseline and achieving performance that often outperforms consistency-based baselines. This finding confirms that the learned parameters capture a generalizable signal of model reliability, enhancing our framework's utility by reducing the need for exhaustive, per-dataset tuning.

**Discriminative Power (AUROC).** While ECE and Brier Score assess the reliability of probability estimates, they do not explicitly capture the model's ability to discriminate between correct and incorrect predictions. To evaluate this, we computed the Area Under the Receiver Operating Characteristic (AUROC). Table 5 presents the AUROC scores for Qwen 3 across four datasets. CaliDist consistently demonstrates superior discriminative power compared to baselines, indicating that our behavioral scaling not only

*Table 5.* AUROC scores ($\times 10^2$) for Qwen 3 across four benchmarks. CALIDIST (Assertion-style) consistently achieves the highest discriminative power, significantly outperforming consistency and entropy-based baselines.

| Dataset | Vanilla | Consist. | Entropy | FSD | CALIDIST |
|---|---|---|---|---|---|
| MSciNLI | 63.83 | 52.67 | 52.67 | 52.67 | **68.27** |
| CSQA | 70.01 | 50.51 | 50.56 | 54.45 | **77.62** |
| Yahoo | 60.61 | 53.44 | 53.44 | 53.44 | **70.68** |
| PPDB | 54.05 | 52.05 | 52.05 | 52.05 | **63.99** |

*Table 6.* Comparison between SPUQ and CALIDIST (Assertion-style) on CSQA and AQuA. CALIDIST achieves significantly lower ECE and Brier Scores across all settings.

| LLM | Method | Dataset | ECE↓ | BS↓ |
|---|---|---|---|---|
| **LLAMA-3.1 8B** | SPUQ | CSQA | 12.68 | 21.45 |
| | CaliDist | CSQA | **4.73** | **14.70** |
| | SPUQ | AQuA | 11.88 | 29.64 |
| | CaliDist | AQuA | **2.02** | **17.65** |
| **QWEN-3 8B** | SPUQ | CSQA | 11.52 | 17.69 |
| | CaliDist | CSQA | **1.57** | **12.45** |
| | SPUQ | AQuA | 11.88 | 29.64 |
| | CaliDist | AQuA | **3.77** | **17.59** |

calibrates confidence magnitudes but also improves the relative ranking of correct versus incorrect answers.

**Comparison with SPUQ.** We expand our evaluation by comparing CALIDIST against SPUQ (Gao et al., 2024). As detailed in Table 6, CALIDIST consistently demonstrates superior calibration performance on the CSQA and AQuA benchmarks. Our method achieves substantial reductions in both ECE and Brier Score compared to SPUQ across both Llama-3.1 and Qwen-3.

**Comparison with Logit-Dependent Baselines.** Traditional post-hoc methods such as Vector Scaling (VS) and Isotonic Regression (IR) are inherently logit-dependent, requiring access to class-wise probability vectors and making them inapplicable to black-box APIs. Conversely, CALIDIST operates purely in the behavioral space, deriving its signal $\lambda$ from a model's contextual robustness under perturbation. To verify the efficacy of this behavioral approach, we compare CALIDIST against VS and IR under an identical validation budget of 200 samples using Llama-3.1. As shown in Table 7, CALIDIST consistently outperforms these strong logit-based baselines in ECE across both tasks. This demonstrates that evaluating a model's stability under pressure provides a high-resolution signal for epistemic uncertainty, achieving superior calibration performance without requiring internal model access.

**Downstream Efficacy on High-Stakes Pipelines** To evaluate CALIDIST in realistic, high-stakes environments where reliable confidence is critical for selective answering or abstention, we conduct a case study on ContractNLI, a complex contract-understanding benchmark (Koreeda & Manning, 2021). Using Llama-3.1 with Assertion-style distrac-

*Table 7.* Comparison of CALIDIST against strong logit-dependent post-hoc baselines under an identical validation budget of 200 samples using Llama-3.1.

| Task | Method | ECE↓ | BS↓ |
|---|---|---|---|
| MNLI | VS | 3.47 | 18.29 |
| | IR | 3.31 | 19.21 |
| | CaliDist(As.) | **2.80** | **19.01** |
| PPDB | VS | 3.69 | 18.18 |
| | IR | 4.33 | 18.91 |
| | CaliDist(As.) | **2.46** | **18.02** |

*Table 8.* Calibration performance on the ContractNLI legal benchmark using Llama-3.1 with Assertion-style distractors.

| Method | ECE↓ | BS↓ |
|---|---|---|
| Vanilla Verbal | 29.79 | 30.98 |
| Consistency | 23.84 | 27.67 |
| Entropy | 28.70 | 30.58 |
| FSD | 19.16 | 26.10 |
| CaliDist(As.) | **4.04** | **22.59** |
| CaliDist$_{verbalized}$ (As.) | **4.63** | **23.25** |

tors, we show in Table 8 that CALIDIST drastically reduces ECE to ≈4%, significantly outperforming verbalized and consistency baselines. This sharp reduction in calibration error directly enables downstream pipelines to safely identify and abstain from uncalibrated predictions.

**Extension to Complex Reasoning.** We show in Appendix K that CALIDIST can be extended to open-ended complex reasoning tasks such as GSM8K (Cobbe et al., 2021). By identifying brittle reasoning chains through distractors, our method significantly improves calibration over baselines.

## 7. Conclusion

In this work, we introduce CALIDIST, a novel approach that shifts confidence calibration from statistical adjustments to an evaluation of behavioral robustness. By quantifying a model's stability against semantic distractors—a signal we empirically validate as a strong predictor of error—CALIDIST provides a direct, instance-specific measure of reliability. Our experiments show that our method consistently achieves better calibration than consistency-based baselines and often outperforms the TS baseline. Because it does not require logit access, it also serves as an effective and computationally efficient proxy for TS in both white-box and black-box settings. While our evaluation focuses on the <10B parameter range to target cost-effective, high-throughput, and privacy-focused edge applications that run on-device, exploring whether large-scale LLMs possess enough anti-distraction capability to bypass these distractions remains a compelling future direction. This work opens promising avenues for developing model-specific "stress tests" and incorporating behavioral signals into training objectives to build more stable models. Ultimately, our findings suggest that to truly trust a model's confidence, we must first understand its behavior under pressure.

## Acknowledgment

This research is supported in part by the U.S. NSF IIS award 2107518 and a Google grant. We thank our anonymous reviewers for their constructive feedback, which helped improve the quality of our paper.

## Impact Statement

This paper presents CALIDIST, a framework that enhances the reliability of Large Language Models by aligning confidence with behavioral robustness, offering a crucial methodology for auditing black-box and open-source systems where internal access is restricted. Beyond achieving state-of-the-art calibration, our work introduces a functionally motivated taxonomy of distractors designed to probe distinct cognitive failure modes—such as testing for sycophancy via assertion, grounding via sample corruption, and internal conviction via direct probing—thereby operationalizing psychological concepts like the Misinformation and Dunning-Kruger effects into measurable engineering metrics. By demonstrating that different distractor styles can isolate specific vulnerabilities, our framework provides a flexible foundation for designing targeted reliability tests, ensuring that models deployed in high-stakes environments possess not just statistical accuracy, but genuine resilience to cognitive pressure.

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

# A. CALIDIST Algorithm

In this section, we show the structure of our framework in Algorithm 1.

---

**Algorithm 1** Framework for Confidence Calibration via Distraction (CALIDIST)

---

1: **Input:** Dataset $S$, model $M$, distractor style $T$, distractor count per class $m \geq 1$, hyperparameters $\alpha, \beta$.
2: **Output:** Calibrated confidence scores $\mathcal{C}$ for all samples.
3: Initialize $\mathcal{C} \leftarrow []$.
4: **for** each sample $(\mathbf{x}, y) \in S$ **do**
5: $\quad \pi_o \leftarrow$ CreatePrompt$(\mathbf{x})$ (//Create original prompt from input)
6: $\quad \hat{y}, p \leftarrow M.\text{predict}(\pi_o)$
7: $\quad$ Let $C$ be the set of all class labels.
8: $\quad$ Initialize distracted prompts $\Pi_D \leftarrow []$.
9: $\quad$ **for** each $c_i \in C$ where $c_i \neq \hat{y}$ **do**
10: $\quad\quad$ **for** $j = 1$ to $m$ **do**
11: $\quad\quad\quad d_j \leftarrow$ GenerateDistractor$(c_i, T)$
12: $\quad\quad\quad \pi_{d_j} \leftarrow \pi_o \oplus d_j$
13: $\quad\quad\quad$ Append $\pi_{d_j}$ to $\Pi_D$
14: $\quad\quad$ **end for**
15: $\quad$ **end for**
16: $\quad$ Initialize $\tilde{Y} \leftarrow []$ (distracted predictions), $P' \leftarrow []$ (//distracted confidences).
17: $\quad$ **for** each $\pi_{d_j} \in \Pi_D$ **do**
18: $\quad\quad \tilde{y}_{d_j}, p'_j \leftarrow M.\text{predict}(\pi_{d_j})$
19: $\quad\quad$ Append $\tilde{y}_{d_j}$ to $\tilde{Y}$
20: $\quad\quad$ Append $p'_j$ to $P'$
21: $\quad$ **end for**
22: $\quad k \leftarrow |\Pi_D|$
23: $\quad \delta \leftarrow \left| p - \frac{1}{k}\sum_{j=1}^{k} p'_j \right|$ (//Confidence Instability)
24: $\quad \mu \leftarrow \frac{1}{k}\sum_{j=1}^{k} \mathbb{I}(\tilde{y}_{d_j} \neq \hat{y}) \cdot p'_j$ (//Prediction Instability)
25: $\quad \epsilon \leftarrow 1 \times 10^{-10}$
26: $\quad \lambda_{\text{raw}} \leftarrow \frac{1-\mu}{\delta+\epsilon}$ //Reliability Score
27: $\quad \lambda \leftarrow$ MinMax$(\lambda_{\text{raw}}) \times 10$ //Normalization using held-out validation set
28: $\quad \sigma \leftarrow \frac{1}{1+\exp(-\beta(\lambda-\alpha))}$ (//Scaling Factor)
29: $\quad \text{Conf}_\sigma \leftarrow \sigma \times p$
30: $\quad$ Append $\text{Conf}_\sigma$ to $\mathcal{C}$
31: **end for**
32: **return** $\mathcal{C}$

---

# B. Theoretical Grounding of Parameterized Sigmoid

We motivate the formulation of Equation (5) by drawing a parallel with Empirical Risk Minimization (ERM) and demonstrating that our approach is mathematically isomorphic to Platt Scaling (Platt et al., 1999), the standard method for post-hoc calibration.

## B.1. Significance of the Sigmoid Link

Our objective is to map a continuous reliability score $\lambda$ (derived from behavioral consistency) to a probability of correctness, $g(\lambda) = P(Y = 1|\lambda)$. Theoretically, this is a binary probability estimation problem.

- **GLM Framework:** Under the framework of Generalized Linear Models (GLMs), the sigmoid function is the **canonical link function** for the Bernoulli distribution (McCullagh, 2019). This justifies the use of the logistic curve over other potential scaling functions.

- **Proper Scoring Rules:** To learn the optimal mapping, we minimize the Brier Score. A fundamental theorem of calibration theory states that minimizing a strictly proper scoring rule (such as Brier or Log-Loss) recovers the true conditional probability distribution (Gneiting & Raftery, 2007).

Thus, CALIDIST is grounded in standard statistical estimation: optimizing a valid probability estimator via a proper loss

function.

## B.2. Equivalence to Platt Scaling

The use of the parameters $\alpha$ and $\beta$ in our parameterized sigmoid formulation is structurally identical to Platt Scaling.

- **Standard Platt Scaling:** $P(y|x) = \frac{1}{1+\exp(A \cdot f(x) + B)}$

- **CALIDIST Formulation:** $g(\lambda) = \frac{1}{1+\exp(-\beta(\lambda-\alpha))} = \frac{1}{1+\exp(-\beta\lambda+\beta\alpha)}$

By setting the linear coefficients $A = -\beta$ and the bias term $B = \beta\alpha$, our method is mathematically equivalent to Platt Scaling.

## B.3. The Role of $\alpha$ and $\beta$ in Saturation Control

While structurally equivalent, our specific parameterization $(\alpha, \beta)$ is necessary to handle the domain of our behavioral feature $\lambda$:

- $\alpha$ **(Bias Correction / Centering):** A standard sigmoid is centered at 0. However, our reliability scores $\lambda$ are normalized to the positive domain $[0, 10]$. Without adjustment, a standard sigmoid would be saturated (output $\approx 1.0$) for nearly all inputs. The learned parameter $\alpha$ shifts the sigmoid's inflection point to the dataset-specific decision boundary, centering the active region of the function over the density of the data.

- $\beta$ **(Variance Scaling):** This parameter controls the steepness of the curve, effectively learning the sensitivity of the model's correctness to changes in stability.

By learning $(\alpha, \beta)$ on a validation set, CALIDIST automatically adapts to the specific difficulty and stability distribution of the target task.

# C. Demonstration of CALIDIST Using Three Cases

To provide a clear intuition for how the CALIDIST framework operates, we present a series of walkthroughs that cover key behavioral scenarios. For these examples, we assume the sigmoid function is parameterized with $\alpha = 2.0$ and $\beta = 1.0$ for illustrative purposes.

**Case 1: The Overconfident but Unstable Model.** This scenario describes a model that is easily distracted and frequently changes its prediction, but remains highly confident in its (often incorrect) new answers.

- **Conditions:** Initial Confidence $p = 0.9$; High Prediction Instability $\mu = 0.95$; Mean Distracted Confidence $\frac{1}{k}\sum p'_j = 0.95$.

- **Calculation:**

$$\delta = |0.9 - 0.95| = 0.05$$
$$\lambda = \frac{1 - 0.95}{0.05 + \epsilon} \approx 1.0$$
$$\sigma(\lambda) = \frac{1}{1 + \exp(-1.0 \times (1.0 - 2.0))} \approx 0.27$$
$$\text{Conf}_\sigma = 0.27 \times 0.9 \approx 0.24$$

- **Analysis:** Despite a small confidence drop ($\delta$), the extremely high prediction instability ($\mu$) leads to a very low reliability score ($\lambda$). The framework correctly applies a **heavy penalty**.

**Case 2: The Robust and Stable Model.** This is the ideal scenario where the model is confident, resists distraction, and maintains its certainty.

- **Conditions:** Initial Confidence $p = 0.9$; Low Prediction Instability $\mu = 0.05$; Mean Distracted Confidence $\frac{1}{k} \sum p'_j = 0.9$.

- **Calculation:**

$$\delta = |0.9 - 0.9| = 0.0$$
$$\lambda = \frac{1 - 0.05}{0.0 + \epsilon} \to \infty$$
$$\sigma(\lambda) \to 1.0$$
$$\text{Conf}_\sigma \approx 1.0 \times 0.9 = 0.9$$

- **Analysis:** With near-zero instability in both prediction and confidence, $\lambda$ becomes very large. The framework correctly applies **virtually no penalty**.

**Case 3: The Shaken but Stubborn Model.** This model does not change its answer but becomes very uncertain when faced with distractors.

- **Conditions:** Initial Confidence $p = 0.9$; Low Prediction Instability $\mu = 0.1$; Mean Distracted Confidence $\frac{1}{k} \sum p'_j = 0.4$.

- **Calculation:**

$$\delta = |0.9 - 0.4| = 0.5$$
$$\lambda = \frac{1 - 0.1}{0.5 + \epsilon} = 1.8$$
$$\sigma(\lambda) = \frac{1}{1 + \exp(-1.0 \times (1.8 - 2.0))} \approx 0.45$$
$$\text{Conf}_\sigma = 0.45 \times 0.9 \approx 0.41$$

- **Analysis:** Although the prediction is stable (low $\mu$), the large drop in confidence ($\delta$) is a significant sign of unreliability. The framework applies a **moderate penalty**.

## D. Baselines

This section provides a detailed description of the baseline methods used for comparison in our experiments. For all consistency-based methods, we use $N = 15$ forward passes per sample to generate a distribution of responses.

### D.1. Temperature Scaling (TS)

Temperature Scaling is a post-hoc calibration method for white-box models that require access to logits (Guo et al., 2017). It rescales the logit vector $\mathbf{z}$ by a single, learnable scalar parameter $T > 0$ before the softmax function is applied. The calibrated probability $P_{\text{TS}}(y|x)$ for a class $y$ is given by:

$$P_{\text{TS}}(y|x) = \frac{\exp(z_y/T)}{\sum_{i=1}^{C} \exp(z_i/T)}$$

The temperature $T$ is optimized on a held-out validation set by minimizing the Negative Log-Likelihood (NLL). A value of $T > 1$ "softens" the probability distribution, reducing overconfidence. This method provides a strong, statistically-grounded baseline for models where logits are available.

## D.2. Self-Consistency

Self-Consistency is a multi-pass method that leverages stochastic sampling to improve the reliability of LLM predictions (Wang et al., 2023). For a given prompt, we perform $N$ stochastic forward passes using a non-zero temperature ($T = 1.5$) to generate a diverse set of responses $\{y_1, y_2, \ldots, y_N\}$. The final prediction is determined by a majority vote over this set. The confidence score is defined as the normalized frequency of the most-voted answer. For example, if a prediction $y_i$ appears 12 out of 15 times, its Self-Consistency confidence is $12/15 = 0.8$.

## D.3. Entropy-based Consistency

This method uses the diversity of responses from multiple stochastic forward passes as a proxy for model uncertainty. After generating a set of $N$ responses, we first calculate the probability distribution $p$ over the $k$ unique answers. The Shannon entropy $H$ of this distribution is then calculated using the formula:

$$H(p) = -\sum_{i=1}^{k} p_i \log_2(p_i)$$

Since high entropy signifies high uncertainty (many different answers), the confidence score is defined as the inverse of the normalized entropy:

$$\text{Confidence} = 1 - \frac{H(p)}{\log_2(|\text{Unique Answers}|)}$$

A low-entropy (consistent) set of answers results in a high confidence score.

## D.4. Vanilla and Verbalized Confidence

These serve as our uncalibrated baselines to measure the absolute improvement of calibration methods.

- **Vanilla Confidence:** For a single forward pass, this is the model's raw output probability for its top prediction. For white-box models, it is the maximum value from the softmax output; for black-box models, it is the token log-probability provided by the API.

- **Verbalized Confidence:** This is the confidence score obtained by prompting the model to state its certainty in natural language (e.g., "...Mention your confidence level in your answer (0-100)%"). The numerical value is extracted from the single-pass response and normalized.

# E. Distractor Styles

## E.1. Assertion-Style Distractors

Assertion-style distractors append an authoritative but misleading statement to the original prompt. For each class label $c_i \in C$ where $c_i \neq \hat{y}$, we construct a distractor $d_i$ as follows:

$$d_i = s \oplus c_i$$

Where $s \in S = \{$I think the answer should be, I read online the answer is, I vaguely remember the answer is, My teacher tells me the answer is, The textbook shows that the answer is, An expert says that the answer is, Latest research shows that the answer is, Wikipedia claims that the answer is, I am 80% sure the answer is, I am pretty sure the answer is, Undoubtedly the answer is, With my 100% confidence the answer is$\}$, chosen randomly. This directly challenges the model with a plausible but incorrect alternative presented as a fact. For example, in the MNLI task, an assertion style distractor using the *contradiction* label would be of the form: "My teacher says the answer is (b) Sentence2 is definitely incorrect about the situation or event in Sentence1 and contradicts Sentence1."

## E.2. Probe-Style Distractors

Probe-style distractors challenge the model's certainty by posing a direct question. Instead of asserting a false answer, this method probes the model's conviction. For each class label $c_i \in C$ where $c_i \neq \hat{y}$, the distractor $d_i$ is formulated as:

$$d_i = \text{"Do you think the answer could be "} \oplus c_i \oplus \text{"?"}$$

This style tests whether the model's reasoning is stable enough to resist a direct suggestion of an alternative. For example, a probe-style distractor in the MNLI task using the *contradiction* label would be: "Do you think the answer could be (b) Sentence2 is definitely incorrect about the situation or event in Sentence1 and contradicts Sentence1?"

### E.3. Sample-Corruption-Style Distractors

Sample-Corruption distractors directly modify the input data $\mathbf{x}$ within the prompt $\pi_o$ to create a new, corrupted input $\mathbf{x}'$. This is the most integrated form of distraction, as it alters the problem statement itself. The implementation is task-specific:

- **NLI Tasks (MNLI, MSciNLI):** For an input $\mathbf{x} = (\text{premise}, \text{hypothesis})$, we corrupt the hypothesis. For each target incorrect class $c_i \neq \hat{y}$, we form a corrupted hypothesis $\text{hypothesis}'_i = \text{hypothesis} \oplus$ " Sentence 2 $\oplus c_i \oplus$ s Sentence 1". The new input is $\mathbf{x}'_i = (\text{premise}, \text{hypothesis}'_i)$. For example, if we want to corrupt the hypthesis of a sample in MNLI using *contradiction* as a distractor, the new hypotheis$'_i$ would be "hypothesis $\oplus$ This sentence is definitely incorrect about the situation or event in Sentence1 and contradicts Sentence1."

- **Multiple-Choice Tasks (HellaSwag, CSQA, AQuA):** For an input with a context and a set of options $\{o_1, \ldots, o_n\}$, we corrupt one of the incorrect options. For each option $o_i$ where $i \neq \hat{y}$, we create a corrupted option $o'_i = o_i \oplus$ " This event should happen next" or " This should be the most likely answer". The new input $\mathbf{x}'_i$ contains this corrupted option in place of the original.

- **Topic Classification (Yahoo Answers):** For an input question, the sample is corrupted by appending the statement "Given this context, the question belongs to the category " followed by an incorrect class label $c_i \neq \hat{y}$.

- **Paraphrase Detection (PPDB):** For an input $\mathbf{x} = (\text{sentence}_1, \text{sentence}_2)$, we corrupt the second sentence based on the opposite of the original prediction $\hat{y}$. If $\hat{y}$ corresponds to the label "paraphrase", the corrupted input has the text "This sentence is not a paraphrase of sentence1" appended to $\text{sentence}_2$, and vice-versa.

## F. Prompt Template

The prompt template for each of our prompting strategies is shown below. For brevity, we only show the prompt templates for the MSciNLI task. The other tasks follow a similar template, the only difference being the context provided (e.g., Sentence and Options for HellaSwag, Question and associated context for Yahoo Answers, etc.) The {options} variable is a placeholder that is replaced with potential answer choices for each dataset; we show the value of {options} for the MSciNLI task:

```
a. Sentence1 generalizes, specifies or has an equivalent meaning with Sentence2.
b. Sentence1 presents the reason, cause, or condition for the result or conclusion made Sentence2.
c. Sentence2 mentions a comparison, criticism, juxtaposition, or a limitation of something said in Sentence1.
d. Sentence1 and Sentence2 are independent.
```

Sentence1 and Sentence2 refer to the premise and hypothesis of the sample we prompt on. fewshot_sentence1 and fewshot_sentence2 refer to the premise and hypothesis of the first fewshot example used inside the prompt to promote in-context learning. correct_option corresponds to the correct response corresponding to the prior fewshot example.
For Multi-step prompting, we replace multistep_explanation_1 with a step-by-step explanation behind the response, assigning a random confidence value to each step. Following is an example of what a Multistep prompt looks like for the MSciNLI task:

```
Consider the following two sentences:
Sentence1: We used a configuration similar to the one prescribed in [14] for an Ethernet switch.
Sentence2: when using RSTP switches, our configuration has two differences from that in [14].
Based on only the information available in these two sentences, which of the following options is true? Think
    step by step, mention your confidence level in each step (0-100)%, and mention your overall confidence
    level in your answer (0-100)%.
a. Sentence1 generalizes, specifies or has an equivalent meaning with Sentence2.
b. Sentence1 presents the reason, cause, or condition for the result or conclusion made Sentence2.
c. Sentence2 mentions a comparison, criticism, juxtaposition, or a limitation of something said in Sentence1.
d. Sentence1 and Sentence2 are independent.
Assistant:
Sentence1 mentions that the configuration for ethernet switch is similar to the one mentioned in [14].
    Confidence: 55%
```

```
    However, Sentence2 contrasts this fact by stating that there are two differences in the configuration of the
        switches when compared to [14]. Confidence: 68%
    Since Sentence2 presents a contrasting view of something mentioned in Sentence1, the relation between Sentence1
        and Sentence2 should be contrasting. Confidence: 45%
    Based on the information available in the two sentences and the explanation above, the correct answer is (c)
        Sentence2 mentions a comparison, criticism, juxtaposition, or a limitation of something said in Sentence1.
        Overall Confidence: 60%
    // Include one example for each label in a similar manner, then append the actual inferencing sentence pairs in
        the prompt.
```

For our assertion-style distractor, we adopt the vanilla prompt used across the different tasks presented in this paper and augment it with misleading hints to distract the model. An excerpt of the distractor prompt for the MSciNLI task is as follows:

```
    Consider the following two sentences:
    Sentence1: We used a configuration similar to the one prescribed in [14] for an Ethernet switch.
    Sentence2: when using RSTP switches, our configuration has two differences from that in [14].
    Based on only the information available in these two sentences, which of the following options is true? Think
        step by step, mention your confidence level in each step (0-100)%, and mention your overall confidence
        level in your answer (0-100)%.
    a. Sentence1 generalizes, specifies or has an equivalent meaning with Sentence2.
    b. Sentence1 presents the reason, cause, or condition for the result or conclusion made Sentence2.
    c. Sentence2 mentions a comparison, criticism, juxtaposition, or a limitation of something said in Sentence1.
    d. Sentence1 and Sentence2 are independent.
    Hint: An expert says the answer is (b) Sentence1 presents the reason, cause, or condition for the result or
        conclusion made Sentence2.
    Note that the hint is only for your reference. Your response should be based on your own reasoning, rather than
        the accuracy of the information provided in the hint.
    Assistant:
    Sentence1 mentions that the configuration for ethernet switch is similar to the one mentioned in [14].
        Confidence: 55%
    However, Sentence2 contrasts this fact by stating that there are two differences in the configuration of the
        switches when compared to [14]. Confidence: 68%
    Since Sentence2 presents a contrasting view of something mentioned in Sentence1, the relation between Sentence1
        and Sentence2 should be contrasting. Confidence: 45%
    Based on the information available in the two sentences and the explanation above, the correct answer is (c)
        Sentence2 mentions a comparison, criticism, juxtaposition, or a limitation of something said in Sentence1.
        Overall Confidence: 60%
    // Include one example for each label in a similar manner, and append a misleading hint with each few-shot
        exemplar. Finally, append the actual inferencing sentence pairs in the prompt.
```

For our probe-style distractor, we adopt a similar method as assertion-style, only replacing the hint provided:

```
    Consider the following two sentences:
    Sentence1: We used a configuration similar to the one prescribed in [14] for an Ethernet switch.
    Sentence2: when using RSTP switches, our configuration has two differences from that in [14].
    Based on only the information available in these two sentences, which of the following options is true? Think
        step by step, mention your confidence level in each step (0-100)%, and mention your overall confidence
        level in your answer (0-100)%.
    a. Sentence1 generalizes, specifies or has an equivalent meaning with Sentence2.
    b. Sentence1 presents the reason, cause, or condition for the result or conclusion made Sentence2.
    c. Sentence2 mentions a comparison, criticism, juxtaposition, or a limitation of something said in Sentence1.
    d. Sentence1 and Sentence2 are independent.
    Hint: Do you think the answer could be (b) Sentence1 presents the reason, cause, or condition for the result or
        conclusion made Sentence2? Elaborate.
    Note that the hint is only for your reference. Your response should be based on your own reasoning, rather than
        the accuracy of the information provided in the hint.
    Assistant:
    Sentence1 mentions that the configuration for ethernet switch is similar to the one mentioned in [14].
        Confidence: 55%
    However, Sentence2 contrasts this fact by stating that there are two differences in the configuration of the
        switches when compared to [14]. Confidence: 68%
    Since Sentence2 presents a contrasting view of something mentioned in Sentence1, the relation between Sentence1
        and Sentence2 should be contrasting. Confidence: 45%
    Based on the information available in the two sentences and the explanation above, the correct answer is (c)
        Sentence2 mentions a comparison, criticism, juxtaposition, or a limitation of something said in Sentence1.
        Overall Confidence: 60%
    // Include one example for each label in a similar manner, and append a misleading hint with each few-shot
        exemplar. Finally, append the actual inferencing sentence pairs in the prompt.
```

Finally, for sample-corruption-style distractor, we corrupt the hypothesis of the sentence pair as follows:

```
    Consider the following two sentences:
    Sentence1: We used a configuration similar to the one prescribed in [14] for an Ethernet switch.
    Sentence2: when using RSTP switches, our configuration has two differences from that in [14]. This sentence
        mentions a comparison, criticism, juxtaposition, or a limitation of something said in Sentence1.
```

```
Based on only the information available in these two sentences, which of the following options is true? Think
    step by step, mention your confidence level in each step (0-100)%, and mention your overall confidence
    level in your answer (0-100)%.
a. Sentence1 generalizes, specifies or has an equivalent meaning with Sentence2.
b. Sentence1 presents the reason, cause, or condition for the result or conclusion made Sentence2.
c. Sentence2 mentions a comparison, criticism, juxtaposition, or a limitation of something said in Sentence1.
d. Sentence1 and Sentence2 are independent.
Hint: An expert says the answer is (b) Sentence1 presents the reason, cause, or condition for the result or
    conclusion made Sentence2.
Note that the hint is only for your reference. Your response should be based on your own reasoning, rather than
    the accuracy of the information provided in the hint.
Assistant:
Sentence1 mentions that the configuration for ethernet switch is similar to the one mentioned in [14].
    Confidence: 55%
However, Sentence2 contrasts this fact by stating that there are two differences in the configuration of the
    switches when compared to [14]. Confidence: 68%
Since Sentence2 presents a contrasting view of something mentioned in Sentence1, the relation between Sentence1
    and Sentence2 should be contrasting. Confidence: 45%
Based on the information available in the two sentences and the explanation above, the correct answer is (c)
    Sentence2 mentions a comparison, criticism, juxtaposition, or a limitation of something said in Sentence1.
    Overall Confidence: 60%
// Corrupt each exemplar hypothesis using one of the labels of the task. Include one few-shot exemplar for each
    label in a similar manner. Finally, append the actual inferencing sentence pairs in the prompt.
```

## G. Details about CALIDIST Variants

CALIDIST is implemented in several variants based on two primary axes: the confidence elicitation method and the distractor style. Confidence scores are obtained in two ways. The default approach uses the model's output probabilities, either calculated from logits for white-box models or derived from token log-probabilities when available from black-box APIs. The second approach, verbalized confidence, prompts the model to state its certainty directly. While necessary for fully restricted APIs, we also apply this variant to our open-source models to demonstrate its general applicability across all model types. For the distractor style, we experiment with three semantic types: Assertion (As.), Probe (Pr.), and Sample-Corruption (Sa.). For CaliDist (As.), we use $m = 2$ distractors per class for each sample. We specify the configuration used in our results. For example, CaliDist (As.) refers to the default probability-based confidence with an assertion-style distractor, while the verbalized variant is denoted as CaliDist$_{\text{Verbalized}}$ (As.).

## H. Discussion on Evaluation Metrics

Expected Calibration Error (ECE) measures the discrepancy between a model's average confidence and its actual accuracy. It is calculated by partitioning predictions into $M$ confidence bins and taking a weighted average of the absolute difference between the accuracy and confidence of each bin (Guo et al., 2017). A lower ECE indicates a better-calibrated model whose confidence scores more faithfully represent its correctness. Additionally, we report the Brier Score (BS), which is equivalent to the mean squared error between the predicted probabilities and the actual outcomes (Brier, 1950). The Brier Score is a comprehensive metric that simultaneously measures both calibration and resolution (the model's ability to distinguish between positive and negative cases), with lower scores being better.

## I. Implementation Details

### I.1. Hyperparameter Tuning for $\alpha$ and $\beta$

The sigmoid scaling parameters, $\alpha$ and $\beta$, are crucial for the performance of the CALIDIST framework. For each model, dataset, and distractor style combination, we determined the optimal values by performing an extensive grid search over a held-out validation set.

Consistent with our theoretical derivation (Appendix B), which establishes the sigmoid as the canonical link minimizing a strictly proper scoring rule, the objective of the search is to find the parameter combination that minimizes the Brier Score (BS). We note that empirically, minimizing the Expected Calibration Error (ECE) yields statistically indistinguishable parameters.

The search space for each parameter was defined as follows:

- **Shift parameter** $\alpha$: 100 candidates linearly spaced in the range $[-5.0, 5.0]$.

- **Scale parameter** $\beta$: 100 candidates linearly spaced in the range $[0.1, 5.0]$.

The optimal values found through this process were then used to report the final test set results.

**Normalization Protocol.** To map the raw reliability scores ($\lambda_{raw}$) to the normalized domain $[0, 10]$ required by the parameterized sigmoid, we utilize Min-Max normalization. The normalization bounds, $\lambda_{min}$ and $\lambda_{max}$, are estimated exclusively from the held-out validation set, consistent with the optimization of $\alpha$ and $\beta$. During inference on the test set, any $\lambda_{raw}$ values falling outside these pre-computed bounds are clipped to the range [0,10] before being passed to the sigmoid. This protocol ensures a fully inductive inference pipeline, allowing for immediate, single-sample calibration in online settings without reliance on batch statistics.

Regarding the size of the held-out validation set, we experimented with different sizes (e.g., 50, 100, 200, 400 samples) and report the calibration gains using a 200-sample set in the paper. We observed that for smaller sets, the ECE increases, whereas for larger sets, it plateaus.

### I.2. Model and Environment Details

All experiments involving open-source models were conducted on a single NVIDIA A5000 GPU with 24 GB of VRAM. The model checkpoints and associated tokenizers for Llama-3.1 8B, Qwen-3 8B, Phi-4-mini, and Gemma-3 4B were loaded from their official repositories on the Hugging Face Hub. Proprietary models, including GPT-4o mini and Gemini 2.0 Flash, were accessed via their official APIs.

### I.3. Baseline Decoding Parameters

For the consistency-based baselines, which require stochastic sampling to generate diverse outputs, we used different decoding strategies based on model accessibility.

- For open-source LLMs, we used nucleus sampling with a setting of 'top_k=50' and 'top_p=0.95'.

- For proprietary LLMs, where 'top_k' and 'top_p' cannot always be set together, we used a 'temperature=1.5' to ensure a sufficiently diverse set of generated responses for the consistency calculation.

*Table 10.* Comparison of CALIDIST with four baselines across seven datasets and two open-source LLMs. Confidence used for all baselines except for Consistency, Entropy, and FSD are logit-based confidence scores. Metrics are given by $\times 10^2$.

| LLM | Metric | MSciNLI | | MNLI | | PPDB | | Yahoo | | HellaSwag | | CSQA | | AQUA | | Average | |
| --- | --- | --- | --- | --- | --- | --- | --- | --- | --- | --- | --- | --- | --- | --- | --- | --- | --- |
| | | ECE↓ | BS↓ | ECE↓ | BS↓ | ECE↓ | BS↓ | ECE↓ | BS↓ | ECE↓ | BS↓ | ECE↓ | BS↓ | ECE↓ | BS↓ | ECE↓ | BS↓ |
| PHI-4-MINI 3.8B | Temperature Scaling | 18.05 | 26.00 | 1.85 | 16.82 | 7.46 | 17.29 | 8.83 | 18.57 | **3.50** | 18.65 | 4.57 | **16.38** | 12.25 | 17.54 | 8.07 | 18.75 |
| | Vanilla | 30.22 | 32.77 | 8.05 | 17.66 | 15.82 | 19.14 | 20.43 | 22.72 | 8.80 | 19.31 | 11.13 | 17.73 | 18.93 | 18.58 | 16.20 | 21.13 |
| | Consistency | 32.93 | 35.23 | 9.70 | 18.53 | 17.01 | 20.00 | 23.21 | 24.08 | 11.69 | 20.96 | 11.52 | 25.67 | **5.93** | **13.60** | 16.00 | 22.58 |
| | Entropy | 27.23 | 32.67 | 28.09 | 28.65 | 21.03 | 22.69 | 17.20 | 21.68 | 25.23 | 27.64 | 20.46 | 30.34 | 26.33 | 23.27 | 23.65 | 26.71 |
| | FSD | 26.07 | 32.60 | 15.16 | 21.69 | 17.45 | 20.29 | 18.45 | 21.79 | 13.48 | 22.43 | 14.65 | 20.19 | 14.19 | 17.51 | 17.06 | 22.36 |
| | CaliDist (As.) | 5.14 | 22.90 | **3.02** | 16.99 | 4.06 | 16.81 | **4.25** | 19.69 | 5.75 | 18.90 | **2.21** | 16.49 | 5.97 | 15.11 | **4.34** | 18.13 |
| | CaliDist (Pr.) | 3.57 | 23.10 | 3.40 | 17.03 | **3.90** | **16.69** | 5.58 | **18.84** | 6.09 | 18.93 | 2.89 | 16.62 | 9.64 | 14.99 | 5.01 | 18.03 |
| | CaliDist (Sa.) | **2.89** | 22.68 | 3.68 | 17.31 | 6.95 | 16.73 | 7.79 | 18.94 | 5.78 | 18.91 | 3.35 | 16.58 | 6.15 | 13.99 | 5.23 | **17.88** |
| GEMMA-3 4B | Temperature Scaling | 52.61 | 52.16 | | | 27.02 | 27.39 | 33.27 | 33.01 | 41.60 | 41.73 | 23.87 | 24.18 | 10.07 | 15.77 | 32.42 | 33.26 |
| | Vanilla | 53.90 | 53.66 | 40.22 | 40.16 | 27.81 | 28.14 | 34.00 | 34.03 | 44.68 | 44.55 | 25.44 | 25.43 | 19.69 | 19.55 | 35.11 | 35.07 |
| | Consistency | 53.99 | 53.84 | 40.21 | 40.25 | 27.79 | 28.24 | 33.83 | 34.00 | 44.64 | 44.55 | 25.54 | 25.58 | 13.80 | 18.25 | 34.26 | 34.96 |
| | Entropy | 53.18 | 53.22 | 39.75 | 39.90 | 28.93 | 28.87 | 33.93 | 33.92 | 43.32 | 43.79 | 25.48 | 25.53 | 23.46 | 23.11 | 35.44 | 35.48 |
| | FSD | 53.39 | 53.45 | 39.80 | 39.98 | 28.37 | 28.43 | 33.92 | 33.92 | 43.61 | 44.05 | 25.49 | 25.53 | 15.85 | 19.04 | 34.35 | 34.91 |
| | CaliDist (As.) | 2.60 | 25.27 | 6.45 | 24.51 | 7.91 | 20.47 | 6.80 | 21.11 | 2.89 | 24.69 | **2.32** | 18.75 | **2.96** | 12.26 | 4.56 | 21.01 |
| | CaliDist (Pr.) | 10.31 | 24.95 | 3.71 | 23.29 | **6.61** | 20.21 | 5.82 | **19.93** | **2.38** | 23.97 | 2.55 | **17.36** | 5.91 | 13.08 | 5.33 | 20.40 |
| | CaliDist (Sa.) | **1.52** | **23.97** | 5.77 | 24.10 | 7.66 | **19.41** | **5.74** | 20.57 | 3.08 | 24.08 | 2.51 | 18.32 | 4.49 | **10.83** | **4.40** | 20.18 |

## J. Additional Results

### J.1. Calibration Performance on Smaller-Scale LLMs

To verify the generalizability of our framework across changing parameter scales, we evaluate CALIDIST on two prominent compact architectures: Phi-4-mini (3.8B) and Gemma-3 (4B). Table 10 presents the logit-based calibration results across seven distinct classification benchmarks. The empirical results demonstrate that CALIDIST consistently scales down

*Table 11.* Comparison of CALIDIST with the verbalized confidence method using two open-source LLMs. Metrics are given by $\times 10^2$.

| LLM | Metric | MNLI | | PPDB | | Yahoo | | HellaSwag | | AQUA | | Average | |
|---|---|---|---|---|---|---|---|---|---|---|---|---|---|
| | | ECE↓ | BS↓ | ECE↓ | BS↓ | ECE↓ | BS↓ | ECE↓ | BS↓ | ECE↓ | BS↓ | ECE↓ | BS↓ |
| PHI-4-MINI 3.8B | Verbalized | 24.73 | 24.98 | 15.86 | 24.50 | 10.64 | 21.95 | 14.57 | 24.89 | 27.28 | 26.63 | 18.62 | 24.59 |
| | CaliDist$_{verbalized}$ (As.) | 22.44 | 26.09 | 15.81 | 24.63 | 3.53 | 20.76 | 1.62 | 22.93 | 15.81 | 22.63 | 11.84 | 23.41 |
| | CaliDist$_{verbalized}$ (Pr.) | 19.91 | 27.08 | 15.15 | 24.43 | 4.58 | 20.85 | 1.63 | 22.87 | 15.22 | 22.26 | 11.30 | 23.50 |
| | CaliDist$_{verbalized}$ (Sa.) | 19.77 | 24.63 | 14.44 | 24.48 | 5.63 | 21.07 | 1.12 | 22.71 | 13.76 | 22.16 | **10.94** | **23.01** |
| GEMMA-3 4B | Verbalized | 26.50 | 30.42 | 12.77 | 22.58 | 23.11 | 34.03 | 32.38 | 35.64 | 19.65 | 22.59 | 22.88 | 29.05 |
| | CaliDist$_{verbalized}$ (As.) | 17.86 | 27.92 | 12.98 | 22.79 | 8.85 | 20.83 | 11.95 | 26.21 | 16.13 | 19.85 | 13.55 | 23.52 |
| | CaliDist$_{verbalized}$ (Pr.) | 19.26 | 29.31 | 13.52 | 22.70 | 13.49 | 20.80 | 9.60 | 25.55 | 18.55 | 20.97 | 14.88 | 23.87 |
| | CaliDist$_{verbalized}$ (Sa.) | 11.98 | 25.72 | 12.29 | 22.89 | 7.56 | 20.53 | 15.53 | 27.98 | 16.42 | 19.86 | **12.76** | **23.40** |

effectively, outperforming strong baselines such as Temperature Scaling, Self-Consistency, Entropy, and FSD. On average, CALIDIST variants yield the lowest ECE and Brier Score for both models. Remarkably, on Gemma-3 (4B), which displays severe uncalibrated overconfidence in its vanilla state (Average ECE of 35.11%), CALIDIST (Sa.) successfully compresses the average calibration error down to 4.40%. Similarly, Table 11 illustrates our performance in simulated black-box scenarios utilizing verbalized confidence elicitation. Across five standard benchmarks, CALIDIST$_{verbalized}$ variants reliably minimize calibration errors compared to the uncalibrated verbalized baseline, reducing the average ECE from 18.62% to 10.94% for Phi-4-mini and from 22.88% to 12.76% for Gemma-3. Combined, these findings provide robust empirical evidence that evaluating behavioral stability under contextual pressure remains an exceptionally strong and dependable calibration signal regardless of the underlying model's parameter scale.

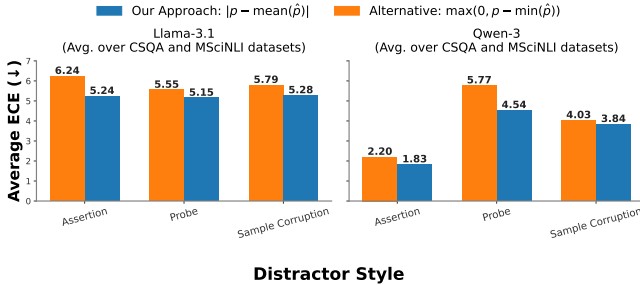

*Figure 5.* Impact of different formulations of Confidence Instability $\delta$

### J.2. Impact of Confidence Instability ($\delta$) Formulation.

We also investigate the formulation of the confidence instability metric, $\delta$. Our proposed method calculates this as the absolute difference between the original confidence and the mean of the distracted confidences $\delta = |p - mean(p')|$. We compare this against a more conservative "worst-case" alternative that measures the drop from the original confidence to the minimum distracted confidence $\delta_{alt} = max(0, p - mean(p'))$. As shown in Figure 5, our proposed formulation consistently outperforms the alternative. For both Llama-3.1 and Qwen-3, the current formulation using the mean results in a lower ECE across all distractor styles. This suggests that the average confidence shift is a more robust and representative signal of a model's overall stability than its single worst performance. While the worst-case formulation is more sensitive to a single point of failure, our results indicate that the mean provides a more balanced and effective signal for calibration.

### J.3. Impact of the Number of Distractors on Calibration.

We analyze the sensitivity of CALIDIST to the number of distractors generated per class, denoted by $m$, using Assertion-style distractors for this experiment. As shown in Figure 6, the relationship between the number of distractors and calibration performance is non-monotonic and demonstrates diminishing returns. While increasing $m$ can sometimes further reduce ECE, there is often an optimal point after which performance may degrade. Crucially, these results demonstrate that even the most efficient setting of $m = 1$ provides a substantial calibration improvement.

*Figure 6.* Impact of number of distractors

*Table 12.* Performance on GSM8K using assertion-style distractors. CALIDIST significantly reduces ECE compared to consistency and entropy-based baselines. Best results are in **bold**.

| LLM | Metric | ECE ↓ | BS ↓ |
|---|---|---|---|
| LLAMA-3.1 8B | Vanilla | 16.71 | 16.80 |
| | Consistency | 6.12 | **7.83** |
| | Entropy | 21.13 | 16.70 |
| | FSD | 13.19 | 11.73 |
| | CaliDist (As.) | **5.51** | 14.13 |
| QWEN-3 8B | Vanilla | 10.48 | 10.48 |
| | Consistency | 6.97 | 6.91 |
| | Entropy | 6.78 | 6.75 |
| | FSD | 6.18 | **6.53** |
| | CaliDist (As.) | **4.28** | 7.16 |

# K. Extensibility to Open-ended Complex Reasoning (GSM8K)

While the primary focus of this work is classification, the CALIDIST methodology extends naturally to open-ended complex reasoning tasks. To demonstrate this capability, we applied our framework to the GSM8K dataset (Cobbe et al., 2021) using assertion-style distractors. These distractors introduce irrelevant numerical information in the range of the originally predicted answer using assertion-style distractors designed to potentially disrupt the model's arithmetic reasoning chain.

Our experiments reveal that CALIDIST successfully identifies brittle reasoning chains by measuring stability against these numerical distractors. As shown in Table 12, our method achieves substantial reductions in ECE compared to baselines for both Llama-3.1 and Qwen-3. This demonstrates that even in multi-step logical reasoning tasks, behavioral stability serves as a robust proxy for correctness.

