# OpenReview forum: "CaliDist: Calibrating Large Language Models via Behavioral Robustness to Distraction"
_ICML.cc/2026/Conference — ICML 2026 regular_

### Official Review · Reviewer_nvkt · 2026-03-02

**Soundness:** 3
**Presentation:** 2
**Significance:** 4
**Originality:** 2
**Overall Recommendation:** 5
**Confidence:** 5

**Summary:**

This paper introduces CaliDist, a post-hoc calibration method for large language models that adjusts token-level or sequence-level predictive distributions to improve confidence calibration without substantially degrading accuracy. The core idea is to model and correct systematic distributional distortions in LLM output probabilities, particularly under distribution shift.

CaliDist estimates calibration parameters based on held-out validation data and applies a transformation to the model’s output distribution at inference time. The authors position the method as lightweight, model-agnostic, and compatible with existing decoding strategies. Empirically, they evaluate across multiple LLMs and tasks, reporting improvements in Expected Calibration Error (ECE), Brier score, and selective prediction metrics, while maintaining competitive accuracy. The paper also studies behavior under distribution shift and varying temperature settings.

**Compliance With Llm Reviewing Policy:**

Affirmed.

**Final Justification:**

Concerns addressed -- I've revised my score accordingly.

**Key Questions For Authors:**

- How does CaliDist compare directly against stronger post-hoc baselines such as vector scaling, Dirichlet calibration, or isotonic regression under identical validation budgets? A rigorous comparison would clarify novelty.

- How sensitive is performance to the size and representativeness of the calibration dataset? If calibration data are mismatched to deployment distribution, does performance degrade sharply?

- Does CaliDist preserve ranking among candidate tokens, or can it alter argmax predictions? Clarifying whether it strictly rescales or can reorder outputs would affect its suitability in certain applications.

- Have you evaluated the method in a real selective-answering or abstention pipeline to demonstrate downstream risk reduction beyond metric improvements?

**Limitations:**

The paper discusses distribution shift and metric-based evaluation, but could more thoroughly address limitations such as dependence on labeled calibration data, interaction with decoding heuristics, and potential instability in low-data regimes.

**Strengths And Weaknesses:**

- The paper addresses a practically important issue: LLM confidence miscalibration, which affects risk-sensitive applications such as selective answering and uncertainty estimation. Improving calibration without retraining large models is a meaningful contribution.

- CaliDist is presented as a simple post-hoc transformation, which increases its potential for adoption. The fact that it operates at the distribution level rather than only via temperature scaling suggests greater flexibility.

- The empirical evaluation includes multiple models and tasks, and reports several complementary calibration metrics rather than relying solely on ECE. The inclusion of selective prediction analysis strengthens the practical framing.

- It is not entirely clear how CaliDist differs theoretically from more expressive variants of temperature scaling or Dirichlet calibration. The paper would benefit from a sharper comparison to strong post-hoc baselines, including vector scaling and isotonic regression.

- The robustness claims under distribution shift are promising but may depend on the specific shifts evaluated. Additional stress tests, such as adversarial or out-of-domain prompts, would clarify the generality of the improvement.

- The computational overhead and interaction with decoding strategies such as beam search or nucleus sampling are only lightly discussed. Since calibration interacts with decoding, more detailed analysis would strengthen the deployment story.

- While the method improves calibration metrics, it is less clear whether this leads to better downstream decision-making performance in realistic pipelines. A task-level case study would enhance the significance argument.

---

> ### Author Rebuttal · Authors · 2026-03-31
>
> We thank the Reviewer for the informative critique of our approach. We address the concerns below and will integrate them into the revision.
>
> > W1, Q1
>
> The primary theoretical distinction between CALIDIST and methods such as Vector Scaling, Isotonic Regression, and Dirichlet Calibration lies in both the input signal and access requirements. These methods are logit-dependent: they remap the model's raw probability distribution using transformations learned on a validation set, and therefore require access to class-wise logit vectors—making them entirely inapplicable to black-box proprietary models. CALIDIST, by contrast, derives its calibration signal from observed behavioral responses to perturbation via the Reliability Score λ, requiring only the ability to query the model. This makes it applicable in white-box, grey-box, and fully black-box settings.
>
> Per the Reviewer's suggestion, we compare CALIDIST against Vector Scaling (VS) and Isotonic Regression (IR) with an identical validation budget of 200 samples on MNLI and PPDB, using Llama-3.1:
>
> | Task | Method        |      ECE |        BS |
> | ---- |:------------- | --------:| ---------:|
> | MNLI | VS            |     3.47 |     18.29 |
> |      | IR            |     3.31 |     19.21 |
> |      | CALIDIST_As | **2.80** | **19.01** |
> | PPDB | VS            |     3.69 |     18.18 |
> |      | IR            |     4.33 |     18.91 |
> |      | CALIDIST_As | **2.46** | **18.02** |
>
> We can observe that CALIDIST outperforms VS and IR despite requiring no access to logits, highlighting its superior calibration performance with the behavioral pivot.
>
> > W2
>
> We clarify that CALIDIST's generalization is already evaluated along two axes: seven benchmarks spanning diverse domains that represent meaningful distributional variation, and the OOD parameter transfer experiment in Figure 4, which directly demonstrates that parameters learned on a source task generalize to a harder target domain without retuning. Regarding adversarial prompts, we note that adversarial perturbations target input robustness—forcing prediction failures—which is distinct from confidence calibration. CALIDIST's distractors are behaviorally motivated probes grounded in the Misinformation Effect and Dunning-Kruger parallels, designed to surface miscalibration rather than induce misclassification. Broader stress testing remains an interesting future direction we will note in the revision.
>
> > W3
>
> We thank the Reviewer for the comment. We refer the Reviewer to `W2` response we provided to Reviewer Zp7f, where we discuss the computational-efficiency of CALIDIST in more detail.
>
> > W4, Q4
>
> We thank the Reviewer for highlighting the importance of evaluating calibration in real-world, high-stakes settings.
> To address this concern, we conducted additional experiments on ContractNLI (Koreeda et al., 2021), a legal-domain benchmark involving contract understanding and entailment. This dataset represents a realistic application scenario in the legal domain, where reliable confidence estimation is critical.
> We use Llama-3.1 with Assertion-style distractors for this experiment. Our results are shown in the following table:
>
> | Method                   | ECE      | BS        |
> |:------------------------ |:-------- |:--------- |
> | Entropy                  | 28.70    | 30.58     |
> | FSD                      | 19.16    | 26.10     |
> | CALIDIST_As          | **4.04** | **22.59** |
> | CALIDIST_As (Verbal) | **4.63** | **23.25** |
>
> As we can observe, CALIDIST substantially outperforms advanced consistency-based baselines. These results demonstrate that CALIDIST remains effective in a real-world legal setting—a scenario with more complex and domain-specific language. We will include these results in the revision.
>
> > Q2
>
> Regarding validation set size, we experimented with different sizes (e.g., 50, 100, 200, 400 samples) and report the calibration gains using a 200-sample set in the paper. We observed that for smaller sets, the ECE increases, whereas for larger sets, it plateaus. We will include this discussion in the revision.
>
> Regarding the degrading performance due to a mismatch in the validation set, we note that in Figure 4, we have shown OOD performance for MSciNLI and AQUA. We observed that OOD models are less calibrated than ID models.
>
> > Q3
>
> *CALIDIST strictly preserves the model's original predictions and cannot alter candidate rankings*. The method operates post-hoc on the label-level confidence of the already-selected argmax prediction—the scaling factor σ is applied multiplicatively to $p$ after $\hat{y}$​ has been determined, making it mathematically incapable of reordering candidates or flipping the top prediction. This ensures that CALIDIST affects only the reliability of expressed confidence, not the underlying decision, making it suitable for high-stakes applications where the model's prediction must remain unchanged. We will clarify this explicitly in the revision.

---

> > ### Author Rebuttal · Reviewer_nvkt · 2026-04-03
> >
> > Concerns addressed -- I've revised my score accordingly.

---

> > > ### Author Response · Authors · 2026-04-06
> > >
> > > We are grateful to the Reviewer for the valuable comments and feedback on our paper, which have helped improve its quality, and we sincerely appreciate the increase in score. We will incorporate all of these updates and improvements into the final version of the paper.

---

### Official Review · Reviewer_Zp7f · 2026-03-11

**Soundness:** 3
**Presentation:** 3
**Significance:** 3
**Originality:** 3
**Overall Recommendation:** 4
**Confidence:** 2

**Summary:**

This paper proposes CALIDIST, a post-hoc calibration method for large language models based on robustness to semantic distractors. The method evaluates prediction and confidence instability under prompt perturbations and uses this signal to adjust model confidence. Experiments on several benchmarks show improvements in calibration metrics compared with existing baselines.

**Compliance With Llm Reviewing Policy:**

Affirmed.

**Final Justification:**

My concerns have been resolved, and I will maintain my current positive score.

**Key Questions For Authors:**

See in the weakness.

**Limitations:**

yes

**Strengths And Weaknesses:**

Strength

Strong empirical performance: Experiments across multiple models and benchmarks show consistent reductions in calibration errors (e.g., ECE) compared with temperature scaling and consistency-based baselines.

Empirical support for the main hypothesis: The analysis shows a clear correlation between prediction instability and model accuracy, suggesting that distractibility is a meaningful reliability signal.

Comprehensive empirical analysis: The paper includes several ablations on calibration mapping, distractor design, and the number of distractors, providing insight into the behavior of the proposed method.

Practical applicability: The approach works in both white-box and black-box settings and does not require additional models or access to internal representations.

Weakness

Dependence on handcrafted distractors: The effectiveness of CALIDIST relies on the quality and diversity of generated distractors. The paper does not fully analyze how sensitive the method is to distractor design or how it generalizes to unseen perturbation styles.

Computational overhead: Generating multiple distractor prompts per sample requires repeated model queries. This could significantly increase inference cost, especially for large API-based models. The paper provides limited discussion of the practical trade-offs.

Limited evaluation beyond classification: Most experiments focus on classification benchmarks. Although the appendix briefly includes GSM8K, broader evaluation on generative tasks (e.g., QA or open-ended reasoning) would strengthen the claims.

---

> ### Author Rebuttal · Authors · 2026-03-31
>
> We are thankful to the Reviewer for the constructive feedback and for appreciating the comprehensive empirical analysis of our approach. We clarify the concerns below and will integrate the discussion in the revision.
>
> > W1
>
> The Reviewer raised a concern about the dependence on handcrafted distractors. We note that we use a handful of 10 templates, e.g., “Wikipedia says”, “My teacher says”, etc., and afterwards the distractors are created by randomly selecting one of these predefined distractor templates in conjunction with the task's label space (as detailed in Algorithm 1 and Appendix D). While we evaluate three distractor styles, our results show consistent calibration improvements across all of them, with an average **70% relative reduction in ECE for LLama-3.1** and **40% for Qwen-3**. This supports the view that CALIDIST leverages a more general signal—prediction instability under perturbation—rather than any particular distractor form. The distractor styles we used are motivated by well-studied behavioral effects (e.g., the Misinformation Effect and the Dunning-Kruger Effect). We emphasize that CALIDIST is not limited to these distractor styles, and other distractors can be designed and integrated into CALIDIST. We agree with the Reviewer that generalization to unseen distractor styles is an interesting avenue for future work.
>
> > W2
>
> In Appendix I.2, we show an analysis that highlights the efficacy of CALIDIST in terms of compute compared to more expensive self-consistency baselines. In particular, the number of forward passes in CALIDIST is determined by the task's label space: $k = (c-1) \times m$. We empirically demonstrate in Appendix I.2 and Figure 6 that even in the most efficient setting of $m=1$ (one assertion-style distractor per unselected class), CALIDIST provides substantial calibration improvements and consistently outperforms consistency baselines that use 15 passes. To provide a more granular view of computational overhead, we show the Average Input-Output Tokens per sample for Llama-3.1 on MSciNLI in the following table:
>
> | Method                 | Total Forward Passes | Output Tokens (Relative) | Input Tokens (Relative) |     ECE |
> |:---------------------- | --------------------:| ------------------------:| -----------------------:| -------:|
> | **Vanilla/Verbalized** |                  $1$ |             $1\times 74$ |          $1\times 1064$ | $25.76$ |
> | **Self-Consistency**   |                 $15$ |            $15\times 74$ |         $15\times 1064$ | $28.05$|
> | **CALIDIST_As (m=1)**  |                  $3$ |            $3 \times 98$ |         $3 \times 1402$ |  $8.02$ |
> | **CALIDIST_Pr**        |                  $3$ |            $3 \times 94$ |         $3 \times 1419$ |  $6.54$ |
> | **CALIDIST_Sa**        |                  $3$ |            $3 \times 65$ |         $3 \times 1184$ |  $6.14$ |
>
> We observe that even though CALIDIST methods require **~5 times fewer tokens** on average, we achieve a **calibration improvement of ~75%** over the more expensive self-consistency methods, highlighting the efficiency and effectiveness of CALIDIST.
>
> > W3
>
> We thank the Reviewer for this comment. We respectfully point out that we **show results on two QA datasets, CSQA and AQUA**, showcasing that CALIDIST works on QA tasks.
> Regarding open-ended tasks, our intent was to demonstrate the promise of CALIDIST via GSM8K results, as evidenced by a substantial improvement in calibration (e.g., an average ~33% relative reduction in ECE on GSM8K compared to the strongest baseline—from 6.15 to 4.13). We agree that a more comprehensive investigation of calibration methods for open-ended tasks is important and necessary. However, given space constraints, incorporating such analyses within the current paper would require compressing the evaluations to make room for the open-ended tasks, which would not do justice to either. We therefore leave a thorough exploration of calibration under distractors in open-ended settings as an important direction for future work.

---

> > ### Author Rebuttal · Reviewer_Zp7f · 2026-04-05
> >
> > My concerns were resolved.

---

> > > ### Author Response · Authors · 2026-04-06
> > >
> > > We are glad that our responses fully clarified the Reviewer’s concerns, as reflected in the Reviewer’s “Fully Resolved” acknowledgment. We are grateful to the Reviewer for the careful evaluation of our paper and for the valuable comments and suggestions, which have helped improve its quality. We will incorporate all of these updates and improvements into the final version of the paper.

---

### Official Review · Reviewer_DucU · 2026-03-12

**Soundness:** 4
**Presentation:** 3
**Significance:** 4
**Originality:** 3
**Overall Recommendation:** 4
**Confidence:** 4

**Summary:**

This paper proposes the method CaliDist to address the issue of confidence calibration in large models. The core breakthrough is to take the robustness of anti-interference behavior as a key dimension for LLM calibration, solving the problems that existing calibration methods rely on internal access to the model and only test internal randomness while ignoring external challenges. CaliDist achieves accurate confidence calibration by quantifying the behavioral stability of LLMs under semantic interference, and is applicable to various white-box/black-box LLMs. It significantly outperforms strong baselines in multiple experiments at a lower cost and can be extended across tasks. The core contributions of this paper are introducing a new behavioral dimension for LLM calibration, proposing a practical and general calibration method, and providing empirical evidence for research on the relationship between behavioral robustness and reliability of LLMs.

**Compliance With Llm Reviewing Policy:**

Affirmed.

**Final Justification:**

After carefully reviewing the authors' rebuttal and acknowledging that my concerns have been fully resolved, I have decided to maintain my overall recommendation (Weak Accept, 4). However, I have increased my Soundness score from 3 to 4 to reflect the clarifications and additional evidence provided by the authors.

**Key Questions For Authors:**

1. Are there automated methods for generating the three types of interference patterns? Are there clear differences in the optimal interference patterns for different tasks? Can the effectiveness of interference be quantified?
2. What are the transfer rules of parameters learned on a certain model across models and tasks? Are there situations where the transfer effect decreases significantly?
3. Can the cost quantification indicators when CaliDist determines parameters be supplemented? And compare with baseline methods to more fully prove the computational efficiency of CaliDist?
4. Is the effect of CaliDist still stable on large-parameter-scale LLMs? Will the increase in model scale improve the anti-interference ability of LLMs, thereby reducing the calibration gain of CaliDist? That is, the strong model capability enables the model to recognize that the problem is the same before and after interference, leading to the failure of interference.
5. When using probe-style questioning for perturbation, the model's confidence itself is prone to change due to the form of the question. Will this phenomenon have a systematic impact on the calibration effect of CaliDist? How to distinguish whether the change in confidence is the real behavioral instability caused by interference or the inherent fluctuation of the model caused by the form of the question?
6. The generalization verification only verifies the transfer from simple tasks to complex tasks. Can it also be transferred across different models?
7. Under different tasks, what are the specific optimal parameter values? Can they be shown?

**Limitations:**

1. Insufficient verification of the method in actual scenarios. The calibration effect of CaliDist has not been verified in actual high-risk application scenarios such as medical diagnosis, financial risk control, and judicial assistance, making it difficult to prove the practicality and reliability of the method in real application scenarios.

**Strengths And Weaknesses:**

Strengths:
1. Reasonable argumentation. It verifies through experiments the strong negative correlation between prediction instability and model accuracy, providing core empirical support for this work. The experiments cover white-box/black-box models, multiple types of NLU tasks, and set up comprehensive baseline methods and ablation experiments.
2. Clear structure and coherent logic. The chapters are naturally connected, the research context is clearly visible, and the formulas and charts are intuitive and standardized.
3. Certain degree of originality. Integrating cognitive psychology into LLM calibration research, it is the first time to take anti-interference robustness as the core dimension of LLM calibration, filling the gap in existing research, and is significantly different from traditional methods that rely on internal access. Testing the anti-interference ability of LLMs from different angles provides an operable scheme for extracting effective behavioral robustness signals, which is different from existing perturbation methods.
4. Certain research significance. It focuses on and solves the core pain points of black-box LLM calibration, without accessing internal model parameters, and has important practical value for high-risk scenarios. At the same time, using cognitive psychology, it pays attention to the phenomenon that high confidence and accuracy are not simply positively correlated.

Weaknesses:
1. The design of the three interference patterns is targeted, but there is still a lack of quantitative analysis of the effectiveness of interference and objective basis for task adaptability, with a certain degree of subjective design.
2. Experiments are only conducted on general scenario NLU datasets, and the task scenarios are too limited to discriminative problems, with insufficient verification on generative and open-ended complex tasks.
3. The current method selects parameters at the task level, but the causes of overconfidence in LLMs are different (some are knowledge errors, some are reasoning defects), and a single task-level parameter cannot adapt to sample-level differences.
4. The baseline comparison system has room for supplementation. It lacks comparison with training-based neural network calibration methods.

---

> ### Author Rebuttal · Authors · 2026-03-31
>
> We thank the Reviewer for the valuable feedback and comments. We address the concerns below and will integrate them in the paper.
>
> > W1,Q1
>
> We would like to clarify that the effectiveness of a distractor is, by design, precisely reflected in the sample-specific values of prediction instability µ (Eq. 1) and confidence instability δ (Eq. 2), which together determine λ (Eq. 4) and consequently the degree to which the confidence is modulated via Eq. 5. This is further corroborated empirically by Figure 1, which shows a strong, consistent negative correlation between the prediction instability µ and predictive accuracy (in other words, the higher the prediction instability, the lower the accuracy).
> To create the distractors, we use a handful of 10 templates, e.g., “Wikipedia says”, “My teacher says”, etc., and afterwards the distractors are created by randomly selecting one of these predefined templates in conjunction with the task's label space (we detail this in Algorithm 1 and Appendix D).
>
> We agree the distractors may appear subjective. However, they are grounded in the Misinformation and Dunning-Kruger Effects, each operationalizing a distinct and documented cognitive failure mode as described in Section 1 and our Impact Statement. In addition, CALIDIST is distractor-agnostic by design—any perturbation eliciting measurable behavioral change serves as a valid calibration signal.
> > W2
>
> We thank the Reviewer for this constructive feedback.  Please refer to our response to weakness `W5` to Reviewer PvKv for further clarification.
> > W3
>
> We agree with the Reviewer that the causes of overconfidence in LLMs are different—some are knowledge errors, some are reasoning defects. This is exactly what CALIDIST captures. We make the following clarifications: Prediction Instability µ and Confidence Instability δ, which are estimated per sample, capture the sample-level knowledge and reasoning errors that the Reviewer refers to. Specifically, knowledge errors typically manifest as prediction instability (high µ), where the model flips its answer under distraction. Reasoning defects, on the other hand, manifest as confidence instability (high δ) even when the prediction remains stable — the "shaken but stubborn" case described in Appendix B.4.
>
> In contrast, α,β are hyperparameters that define the shape of the parametric sigmoid function (Generalized Logistic Function), and are estimated per task/dataset.
> > W4
>
> We performed additional comparisons with vector scaling and isotonic regression and found that CALIDIST outperforms these baselines. Please see our response to `W1,Q1` of Reviewer nvkt for details.
> > Q2, Q6
>
> We thank the Reviewer for the suggestion. We conducted a cross-model parameter-transfer experiment and generally found that calibration performance degraded. We conclude that the primary practical value of CALIDIST lies in its robust cross-task generalizability within a single model (Fig. 4), enabling an efficient "tune-once, deploy-always" solution for diverse real-world applications.
> > Q3
>
> We point the Reviewer to the `W2` response we provided to Reviewer Zp7f, which discusses the computational efficiency of CALIDIST in more detail.
> > Q4
>
> We thank the Reviewer for this question. Our decision to focus on models in the <10B range is motivated by the fact that smaller models are the primary choice for cost-effective, high-throughput, and edge-computing applications that can run efficiently on-device. On-device deployment is also essential for privacy-focused reasons. Nevertheless, extending CALIDIST to large-scale LLMs is an interesting future direction.
> > Q5
>
> We thank the Reviewer for this question, but clarify that the phenomenon described is precisely the signal CALIDIST is designed to capture. As motivated in Section 1, a model with robust knowledge should behave analogously to a confident human expert: it should not materially change its prediction or waver in certainty simply because an alternative is suggested. This is directly grounded in our cognitive psychology parallels — a model with truly robust knowledge should be invariant to the form of the challenge, making any confidence shift itself a meaningful signal of unreliability. The empirical validation in Figure 1 corroborates this: the negative correlation between instability signal µ and accuracy confirms that µ is genuinely predictive of error rather than a systematic artifact of question form. Furthermore, the convergence of strong calibration improvements across all three structurally distinct distractor styles—assertion, probe, and sample-corruption—provides additional evidence that the signal reflects genuine model unreliability rather than a form-specific response bias.
> > Q7
>
> We show the values of the hyperparameters across two tasks (MNLI and MSciNLI) for Qwen-3 using assertion-style distractors in the following table:
>
> | Task    | α| β |
> |:------- | --------:| -------:|
> | MNLI    |   -4.091 |     0.1 |
> | MSciNLI |    3.687 |     0.1 |

---

> > ### Author Rebuttal · Reviewer_DucU · 2026-04-06
> >
> > The authors have provided clear and comprehensive responses to all my concerns. They clarify that distractor effectiveness is quantitatively measured by prediction instability μ and confidence instability δ, which are strongly correlated with accuracy. They distinguish sample-level signals (μ, δ) capturing knowledge errors and reasoning defects from task-level hyperparameters (α, β), solving the problem of sample-level adaptation. Additional comparisons with advanced baselines, cross-task/cross-model transfer analysis, computational cost verification, and legal-domain experiments further improve the paper’s reliability and practicality.

---

> > > ### Author Response · Authors · 2026-04-06
> > >
> > > We are glad that our responses fully clarified the Reviewer’s concerns, as reflected in the Reviewer’s “Fully Resolved” acknowledgment. We are grateful to the Reviewer for the careful evaluation of our paper and for the valuable comments and suggestions, which have helped improve its quality. We will incorporate all of these updates and improvements into the final version of the paper.

---

### Official Review · Reviewer_PvKv · 2026-03-13

**Soundness:** 2
**Presentation:** 3
**Significance:** 2
**Originality:** 3
**Overall Recommendation:** 3
**Confidence:** 3

**Summary:**

This paper proposes CALIDIST, a post-hoc calibration method for classification-style LLM use. For each input, it generates semantic distractors tied to alternative labels, measures prediction instability μ and confidence instability δ under those distractors, combines them into λraw = (1−μ)/(δ+ϵ), normalizes λ to [0,10], maps λ through a learned sigmoid, and rescales the original confidence. The evaluation spans seven NLU benchmarks and six LLMs, with comparisons to vanilla confidence, temperature scaling, and consistency-based baselines. The core empirical claim is that robustness to distractors is a useful sample-level calibration signal.

**Compliance With Llm Reviewing Policy:**

Affirmed.

**Ethical Review Concerns:**

no ethical concern

**Key Questions For Authors:**

see weakness

**Limitations:**

The author did not discuss limitation

**Strengths And Weaknesses:**

Strengths

1. The paper introduces an interesting behavioral signal for calibration. Using resistance to misleading context is a fresh angle relative to global rescaling and self-consistency methods, and the core construction via μ, δ, and λ is easy to understand.

2. The empirical coverage is solid: seven benchmarks, six models, open-source and proprietary settings, log-probability and verbalized confidence, plus ablations on the sigmoid, δ formulation, distractor count, OOD transfer, and a GSM8K extension.

3. The efficiency story is plausible. The number of distracted passes scales as k=(c−1)m, and the ablation suggests even m=1 can already provide substantial gains on some tasks, which is attractive versus 15-pass consistency baselines on small-label problems.

Weaknesses

1.The most serious issue is the under-specified normalization step. Eq. 4 requires MinMax normalization of λraw before the sigmoid, but the paper never states over what population this MinMax is computed, and Algorithm 1 omits the normalization entirely. This directly affects reproducibility and whether the method can be deployed online or per instance.

2. The paper is internally inconsistent about how α and β are fit. The main text says the validation-set search minimizes ECE, whereas Appendix H says it minimizes Brier Score and only notes that ECE gives similar parameters. The reported results are therefore not fully auditable as written.

3. The theoretical grounding is weaker than advertised. Appendix B shows that a sigmoid on λ is structurally equivalent to Platt scaling, but the deployed estimator is not that sigmoid; it is σ(λ,α,β) multiplied by the original confidence p. The theory therefore does not fully justify the actual formula used in Eq. 5.

4. The practical framing is weaker than the paper suggests. The method still needs labeled validation data and a 100×100 grid search for α and β per setting, so it is not a parameter-free plug-in. Even after that fitting, some gains are marginal or absent; for example, in Table 1 Qwen-3 on PPDB with probe-style distractors changes ECE only from 39.46 to 39.12, and in the proprietary verbalized setting Gemini-2.0-Flash on HellaSwag stays at 22.60 ECE across all CaliDist variants.

5. The extension beyond finite-label classification is not yet convincing. The method is naturally defined around alternative class labels via k=(c−1)m, so open-ended use requires a different construction. In Appendix J, CALIDIST improves GSM8K ECE but has worse Brier Score than FSD for both Llama-3.1 (14.22 vs 11.73) and Qwen-3 (7.16 vs 6.53), which weakens the generalization claim.

---

> ### Author Rebuttal · Authors · 2026-03-31
>
> We thank the Reviewer for the constructive feedback and valuable comments. We are encouraged that the Reviewer found our method to be a fresh angle in calibration research with solid empirical coverage. We address the concerns below and will integrate them in the paper.
>
> > W1
>
> We thank the Reviewer for identifying this omission. We apologize for this oversight and clarify that the $min$ and $max$ values for $λ_{raw}$ normalization are computed over the held-out validation set, ensuring the normalization is grounded in the same distribution used to learn the sigmoid parameters α and β. We 'freeze' the $min$ and $max$ values after the validation phase and perform per-instance calibration during inference. We will update the text and the algorithm accordingly.
>
> > W2
>
> We thank the Reviewer for pointing this out. We clarify that the mention of ECE in Section 4 (Hyperparameter Tuning) was a typographical error. The correct optimization objective used for all experiments was the BS, as specified in the theoretical framework of Section 3.3 (Final Confidence Calibration, Line 222) and the implementation details of Appendix H. We will correct the text in Section 4.
>
> > W3
>
> We thank the Reviewer for this insightful observation. Indeed, Appendix B, as currently written, establishes that the sigmoid σ(λ,α,β) is structurally equivalent to Platt Scaling as a probability estimator, but Eq. 5 deploys it as a multiplicative factor applied to the original confidence $p$. Our wording was too strong, and we will revise it accordingly. Our intended claim is that the sigmoid sub-component is theoretically motivated by standard Bernoulli link-function arguments, while the multiplicative combination with $p$ is a modeling choice that preserves the model’s native prediction and uses distraction-derived robustness as a sample-specific trust discount.
>
> Importantly, Eq. 5 can be rewritten in log-space as: log Conf$_σ$ = log $p$ + log σ(λ, α, β). This reveals the formula as an additive correction to the model's log-confidence. Since α and β are learned by minimizing the BS—a strictly proper scoring rule—this correction is optimized via principled ERM, preserving the theoretical justification.
> > W4
>
> We thank the reviewer for this feedback and clarify that our intent is not to claim CALIDIST is parameter-free, but rather that it is a lightweight post-hoc method requiring only two scalar hyperparameters (α,β) estimated on a small validation set. We will update the manuscript to ensure this distinction is explicit and to clarify that the grid search is a one-time offline cost. Regarding the magnitude of improvements, while isolated cases may show marginal gains, the average performance across benchmarks demonstrates a substantial and consistent calibration advantage. On average across all seven datasets in Table 1, CALIDIST achieves a ~57% relative reduction in ECE (from 15.18 to 6.58) and ~13% in BS (from 20.91 to 18.18) for Llama 3.1 and we see similar improvements for Qwen-3. Even in the specific Qwen-3/PPDB case cited, while the "probe" style shows marginal improvement, the "sample-corruption" style on that same model/task reduces ECE by ~35% (from 38.96 to 25.15), highlighting the value of our distractor taxonomy in identifying the specific behavioral stressors to which a given model is susceptible. Additionally, for smaller models such as Phi-4-mini and Gemma-3-4B, we observe similar improvements (as shown in Table 7, Appendix J).
> On proprietary models, we obtain an average ~33.5% relative reduction in ECE across the logprob setting, and while gains in the verbalized setting are more modest, we still observe a ~5% average ECE reduction.
> To better reflect these trends, we will add "Average" columns to our tables in the final paper, as we believe these consistent, significant improvements across diverse architectures demonstrate the practical utility of the framework beyond specific edge cases, and we hope the Reviewer evaluates our submission based on its core contributions rather than the two cases where CALIDIST shows marginal or no gains.
>
> > W5
>
> We thank the Reviewer for this insightful feedback. Our intent in including the open-ended task results—as shown in Table 6, Appendix J—was to demonstrate the promise of CALIDIST, as evidenced by a substantial improvement in calibration (e.g., an average ~33% relative reduction in ECE on GSM8K compared to the strongest baseline—from 6.15 to 4.13). We agree that a more comprehensive investigation of calibration methods for open-ended tasks is important and necessary. However, given space constraints, incorporating such analyses within the current paper would require compressing the evaluations to make room for the open-ended tasks, which would not do justice to either. We therefore leave a thorough exploration of calibration under distractors in open-ended settings as an important direction for future work.

---

> > ### Author Rebuttal · Reviewer_PvKv · 2026-04-04
> >
> > Thank you for the detailed rebuttal

---

> > > ### Author Response · Authors · 2026-04-06
> > >
> > > We are glad that our responses fully clarified the Reviewer’s concerns, as reflected in the Reviewer’s “Fully Resolved” acknowledgment. We are grateful to the Reviewer for the careful evaluation of our paper and for the valuable comments and suggestions, which have helped improve its quality. We will incorporate all of these updates and improvements into the final version of the paper.

---

### Decision · Program_Chairs · 2026-04-30

**Decision:**

Accept (regular)

**Comment:**

The paper introduces a post-hoc calibration method CaliDist for LLMs based on behavioral robustness to semantic distractors, and evaluates it across multiple models and NLU benchmarks. Reviewers generally agree that the paper proposes an interesting and original perspective on calibration, with strong empirical support showing consistent improvements. A number of concerns were initially raised regarding clarity (e.g., normalization details, objective inconsistencies), theoretical justification, reliance on handcrafted distractors, and limited evaluation beyond classification tasks. However, these issues were largely addressed in the rebuttal. While some limitations remain, the overall consensus is that the method is technically sound and offers a useful new signal for calibration. Based on these considerations, I recommend acceptance.